



# Combining observational data and numerical models to obtain a seamless high temporal resolution seasonal cycle of snow and ice mass balance at the MOSAiC Central Observatory

Polona Itkin[1] and Glen E. Liston[2]

[1]UiT The Arctic University of Norway
[2]Cooperative Institute for Research in the Atmosphere, Colorado State University, Fort Collins, CO, USA

**Correspondence:** Polona Itkin (polona.itkin@uit.no)

**Abstract.** Multidisciplinary drifting Observatory for the Study of Arctic Climate (MOSAiC) observations span an entire annual cycle of Arctic snow and sea ice cover. However, the measurements of atmospheric and ocean forcing, as well as distributed measurements of snow and ice properties were occasionally interrupted for logistical reasons. The most prolonged interruption happened during onset of the summer melt season. Here we introduce and apply a novel data-model fusion system that can assimilate relevant observational data in a collection of modeling tools (SnowModel-LG and HIGHTSI) to provide continuous high temporal resolution (3-hourly) time series of snow and sea ice parameters over the entire annual cycle. We used this system to analyze differences between the three main ice types found in the MOSAiC Central Observatory: relatively deformed second year ice, second year ice with extensive smooth refrozen melt pond surfaces, and first year ice. Since SnowModel-LG and HIGHTSI were used in a 1-D configuration, we used a sea ice dynamics term $D$ to parameterize the redistribution of snow to newly created ridges and leads. $D$ correlated highly with the sea ice deformation ($R^2$=59%, N=33) in the vicinity of the observatory and was at times as high as 10% of all winter snowfall. In addition, we show, in separate simulations for level ice, that snow bedforms with thin snow in the bedform troughs largely control the ice growth. Here, mean snow depth minus one standard deviation was required to simulate realistic sea ice thickness using HIGHTSI; we surmise that this is accounting for the control of relatively thin snow on local ice growth. Despite different initial sea ice thickness and freeze-up dates, sea ice thickness of level ice across all ice types became similar by early winter. Our simulations suggest that the mean (spatially distributed) MOSAiC snow melt onset began in late May, but was interrupted by a snowfall event and was delayed by 3 weeks until mid June. The level ice started to melt in the last week of June. Depending on the sea ice topography, the ice was snow-free by late June and early July.



## 1 Introduction

Sea ice is an important regulator of the Arctic Ocean energy budget. Sea ice limits the transfer of energy (McPhee, 2012) and light (Arrigo et al., 2012) from the atmosphere to the ocean, and constrains the transfer of heat from ocean to atmosphere (Maykut and Untersteiner, 1971). A strongly controlling component of this flux-dampening effect of sea ice is associated with presence, quantity, and physical properties of snow that may cover it. Snow on sea ice is the main regulator of level sea ice thickness (Sturm et al., 2002b; Perovich et al., 2011; Itkin et al., 2023). Snow has an order of magnitude lower thermal conductivity than sea ice and, in winter, inhibits ice growth by insulating it from the relatively cold atmosphere (Maykut and Untersteiner, 1971; Sturm et al., 2002b). In summer, snow slows down sea ice melting by increasing the albedo (Perovich et al., 2011). These roles of snow governing sea ice evolution has become more evident with the general ice cover thinning taking place in association with recent climate change (Meredith et al., 2019). Now, both first-year ice (FYI) and second-year ice (SYI) can reach similar thicknesses at the end of the winter ice-growth season (Itkin et al., 2023), and can reach similar melt pond distributions in summer (Webster et al., 2022).

Since the density and thermal properties of sea ice and snow are so different, both need to be accounted for in order to understand the state of the coupled snow-sea ice system. This remains one of the major challenges of satellite remote sensing of snow and sea ice in polar regions and, consequently, impacts the validation of climate models that rely on these data (Gerland et al., 2019). The drift of the Multidisciplinary drifting Observatory for the Study of Arctic Climate (MOSAiC), 2019-2020, was the largest research expedition to the Arctic Ocean to date (Figure 1). It was implemented to help fulfill this need and others, and had a key goal of collecting data relevant for climate process studies across the entire snow and sea ice annual cycle, and to do so with a special focus on the less-studied winter period (Nicolaus et al., 2022).

MOSAiC collected an unprecedented quantity of high spatial resolution snow and sea ice property data over snow and ice covers of various ages (Macfarlane et al., 2023b; Itkin et al., 2023; Oggier et al., 2023b,a; Raphael et al., 2024). Sea ice thickness and snow depth measurements were collected along transect lines approximately 1-kilometer long with measurements made approximately every 7 days during the entire annual cycle of Arctic sea ice cover from freeze up, through winter, and until the end of melt (Itkin et al., 2023). In addition, a system of distributed snow pit measurements were used to measure snow density and other snow properties approximately every 7 days throughout the year (Macfarlane et al., 2023b). While the measurements were generally collected once a week, early fall MOSAiC ship arrival, late freeze up of the melt ponds and FYI, sporadic weather, sea ice deformation events, crew exchanges, and summer ship departure, caused discontinuities in the observation time series of up to 2.5 months. These discontinuities in the observation data time series (generally summarized as late onset, winter and spring fragmentation, and early truncation), limits its use for process studies and upscaling for satellite remote sensing and numerical model applications.

In this study, we use atmosphere, ocean, ice, and snow measurements from the MOSAiC Central Observatory (Itkin et al., 2023; Cox et al., 2023; Matrosov et al., 2022; Macfarlane et al., 2023b), and autonomous instrument measurements in the surrounding MOSAiC Distributed Network (Lei et al., 2022; Benjamin Rabe, 2024), as forcing and assimilation data to drive a collection of one-dimensional, physics-based, data assimilation, mass balance, and thermodynamic snow and sea ice models





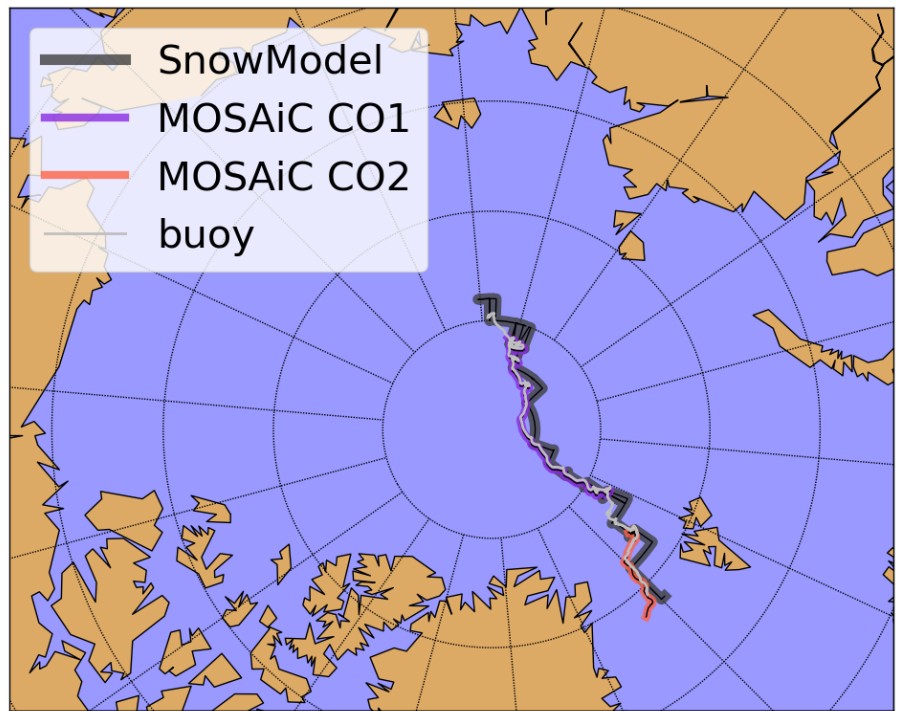

**Figure 1.** The drift of the Multidisciplinary drifting Observatory for the Study of Arctic Climate (MOSAiC) from October 2019 through July 2020. The first observatory (CO1) lasted from October to breakup in May. The second observatory (CO2) was established in June and July. Autonomous buoys (i.e., drifters) recorded the entire drift. The drift path was extended back to August 2019 using 25-km spatial resolution ice motion vectors from the National Snow and Ice Data Center.

as part of a process we call data-model fusion (*sensu* Boelman et al., 2019; Reinking et al., 2022). This data-model fusion
methodology was used to fill in multi-week gaps in the MOSAiC snow and ice data time series and increase their observational
data frequency from weekly to temporally continuous (3-hour time step) covering the full annual cycle.

In addition, previous analyses of MOSAiC snow depth data (Itkin et al., 2023) showed that snow depth on level ice accumu-
lated slower than on deformed ice, and that occasional decreases in mean snow depth were observed over all sea ice types. The
snow and sea ice thicknesses over various ice ages at MOSAiC were similar at the end of the winter. Here, we use physics-
based snow and ice models to explain this snow and ice mass balance evolution during the winter snow and ice accumulation
period and those during the spring and summer snow and ice melt period.

The two basic assumptions of this work are 1) that the spatially and temporally distributed data collected at MOSAiC are of
excellent quality and represent the general annual snow and sea ice cycles, and 2) that the numerical modeling tools are able to
simulate all of the important first-order climate processes.



## 2   MOSAiC observation summary

### 2.1   Sea ice types at MOSAiC

The sea ice cover in the MOSAiC Central Observatory, surrounding *RV Polarstern*, was mainly composed of three ice types:

1. Predominantly **deformed SYI** that survived the summer melt. This ice type had very few level ice surfaces. The deformed ice (rubble and ridges) was consolidated during the summer melt period into hummocks and old ridges. The ponds that developed on this ice type over summer, remained fresh and were not connected to the sea water.

2. Predominantly level and **ponded SYI**. At the beginning of October, this ice had very little snow cover and bare surfaces of refrozen melt ponds were visible. This implies that no snow had accumulated on it during summer. Many of the refrozen melt ponds had previously been connected to the ocean and had a salty surface (Macfarlane et al., 2023b).

3. Predominantly **level FYI** that was still forming throughout October (Itkin et al., 2023). As the thinnest and most level ice type, this ice underwent frequent deformation and was challenging to sample.

The exact spatial distribution of these three ice types is not known, but based on the ground measurements, visual observations, airborne maps, and satellite images, we reconstructed the ice chart of the MOSAiC observatory shown in Figure 2.

### 2.2   Observation transects

Snow depth and sea ice thickness data collected along repeated transect lines at MOSAiC were designed to cover diverse ice surfaces that represented large areas of different snow and ice characteristics (Itkin et al., 2023). These repeated, long transect measurements provided statistically significant snow and ice property datasets over each of the three main MOSAiC ice types, and were specifically prioritized over point-wise, but temporally continuous, sampling provided by individual or clusters of autonomous instruments as in, e.g., Lei et al. (2022), Perovich et al. (2023), Raphael et al. (2024), and Salganik et al. (2023b). They are also superior to large-scale helicopter transects that are not generally repeated over the same sea ice, can be sporadic, and cannot distinguish between snow and sea ice thickness (von Albedyll et al., 2022).

The transects were sampled over all three ice types listed in the section above. These transects are called Nloop, Sloop, and Runway (Figure 2):

1. **Nloop**: representing predominantly **deformed SYI**. This transect was approximately 1.5 km long and sampled between the second half of October 2019 and the beginning of May 2020.

2. **Sloop**: representing predominantly level and **ponded SYI**. This transect was approximately 1.5 km long and sampled between the end of October 2019 and the beginning of May 2020.

3. **Runway**: representing **level FYI**. This transect was approximately 1.0 km long. This transect was sampled only three times in January and February 2020. Afterwards it was not accessible and partially destroyed by ice motion.



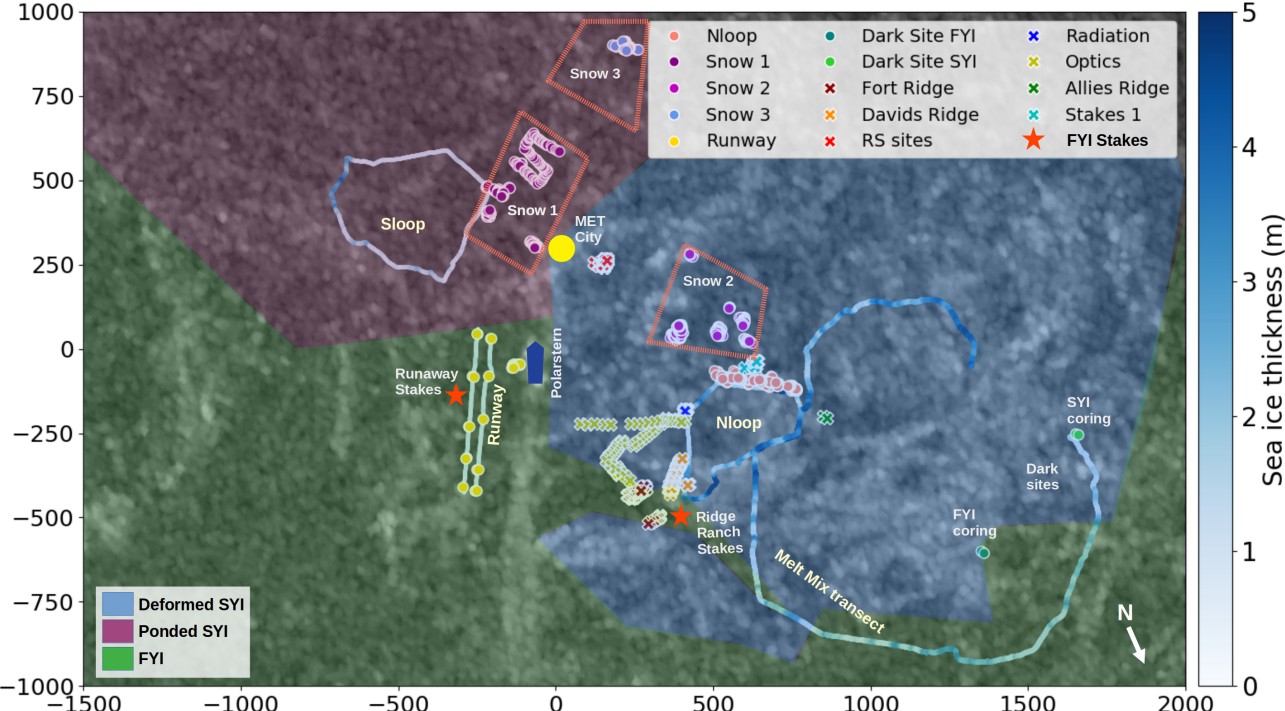

**Figure 2.** Ice chart with three ice types, transect lines, and point observation sites in the MOSAiC Central Observatory. The transect lines and point sites are all drift-corrected to overlay on the observatory location in October 2019. Transect line ice thickness is displayed in blue shades. Individual snow pit point measurements are marked by symbols, and snow pit polygons are outlined by red dash lines. Dark site with ice coring activities, MET City, and RV *Polarstern* location (blue ship-shaped feature at map origin) are also marked. RV *Polarstern* is located at the origin of the map coordinate system (0,0). The size of the map is 3.5 km by 2.0 km with distances (m) from origin marked on the edge of the map. The ship's heading in October 2019 (approximately south-southwest) determines the map orientation. The North arrow shows true North in October 2019. The background is a Radarsat-2 SAR image from 31 December 2019. The brighter features on the Radarsat-2 image correspond to deformed ice and SYI. Darker features are level SYI and FYI.

In May the sampling along these winter-time transects was discontinued and in mid-June a new **Melt Mix transect** (Figure
2) was established. Part of Nloop and the surrounding FYI were integrated into this new line, but the majority of the ice surfaces were deformed ice. This transect was approximately 3.0 km long and represented all ice types.

## 2.3  Point observation sites

In addition to transects, this paper relies on ice thickness, snow depth, snow structure, sea ice deformation, ocean heat fluxes, and meteorological data measured at individual point observation sites in the MOSAiC Central Observatory and surrounding
Distributed Network (Nicolaus et al., 2022; Benjamin Rabe, 2024).





The Dark Sites ('Dark Site SYI' and 'Dark Site FYI', Figure 2) were ice coring sites where ice thickness and snow depth were measured regularly between October and July (Evgenii Slaganik, personal communication and Oggier et al. (2023b,a)). The location of these sites was chosen on level SYI and FYI away from the ship to avoid any light or chemical pollution. The FYI at the Dark Site was formed during freeze-up around 1 October and, as such, was a few weeks older than the Runway FYI.

Ice coring is a destructive measurement, so the same ice can only be sampled once and the next sample is taken a few meters away. Although only level ice was supposed to be sampled, some samples contained deformed ice that transiently increased the measure of sea ice thickness. At these sites, the ice surface was level and the snow depth was not affected by the vicinity of ridges, yet the number of samples were small and exhibit some variability due to snow bedforms.

Also, repeated (non-destructive) measurements over exactly the same snow and ice were read at snow stake and ice hot wire

clusters (Raphael et al., 2024). These clusters were relatively small (approximately 10 stakes); herein we only use two stake clusters on the **level FYI** to augment the low quantity of transect data over this ice type. The two stake sites we used were 'Ridge Ranch Stakes' deployed in January close to the Fort Ridge, and 'RunAway Stakes' deployed in February at the Runway (Figure 2).

Inside the MOSAiC Central Observatory, as well in the Distributed Network surrounding it, 210 autonomous buoys (drifters)

were deployed (Benjamin Rabe, 2024; Bliss et al., 2023). While the majority of these buoys were deployed on level sea ice that was stable at their deployment time in October (thick SYI), the sea ice between them was composed of all three ice types described in the section above. A subset of these buoys, 4 buoys transmitting a GPS signal throughout October to May, and forming a square with sides of approximately 5 km (Figure 3), was used to estimate sea ice deformation. Also a selection of 23 buoys within the radius of about 15 km from the RV *Polarstern*, equipped with thermistor chain measuring snow and sea ice

temperature, were used to estimate heat fluxes at the ice-ocean interface (Lei et al., 2022). This network included three L-Sites that were 10 to 20 km from the Central Observatory, where some snow pits were dug with assistance of the helicopter landings throughout the MOSAiC drift (Macfarlane et al., 2023b).

The snow pit sampling sites at MOSAiC were distributed over all three sea ice types (Figure 2; Macfarlane et al. (2023b)). Within these ice types, the majority of the snow pits were dug and measured on the level ice, while some sites were in ridges.

Similar to ice coring, these measurements are destructive and every next pit was dug meters away from the old one and potentially in a different relative position inside a snow bedform. The snow pit locations representative for each of the ice types in the section above were:

1. **Deformed SYI**: Nloop, Dark Site SYI, Fort Ridge, David's Ridge, Allies Ridge, and L Sites.

2. **Level and ponded SYI**: Snow 1, Snow 2, Snow 3, and RS Sites.

3. **Level FYI**: Runway, Stakes 1, Dark Site FYI, and partially Optics and Radiation.

Temperature, relative humidity, and wind observations used in this paper were collected at METCity on a meteorological tower (Figure 2) at 10 m height (Cox et al., 2023). Precipitation was measured by the precipitation radar on RV *Polarstern* (Matrosov et al., 2022).



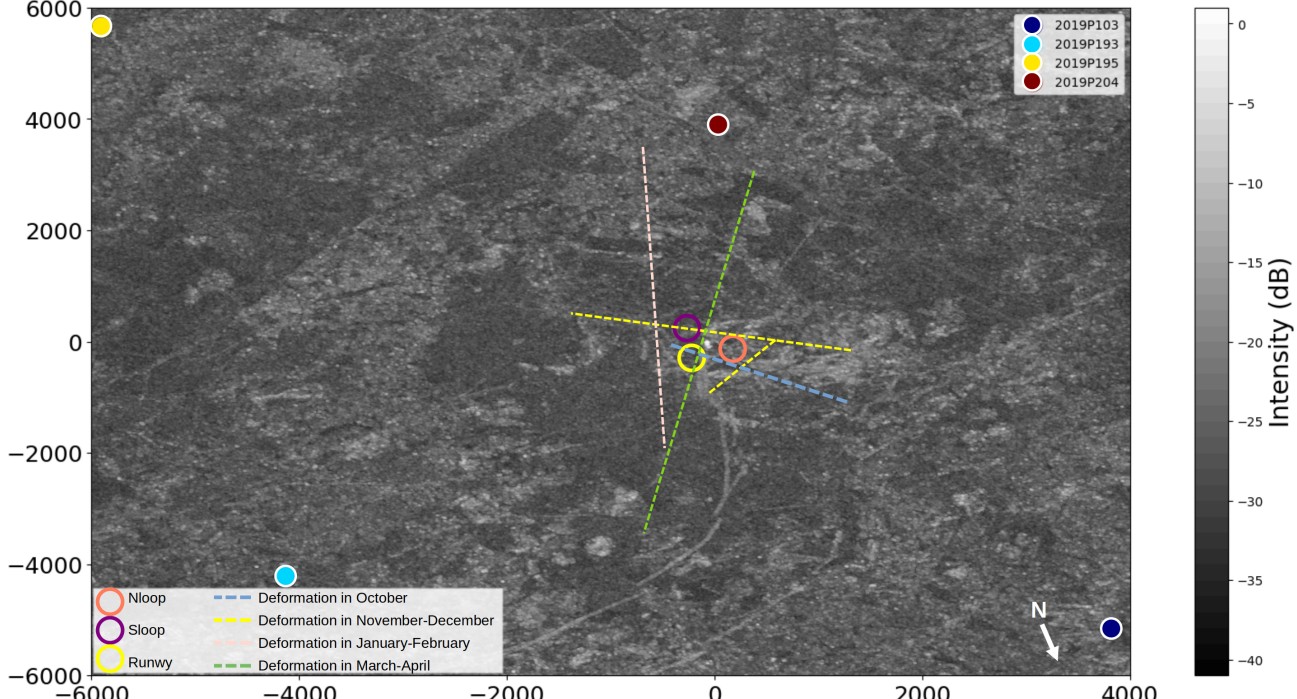

**Figure 3.** Broader MOSAiC area sea ice cover, GPS buoys, and deformation zones in the vicinity of the Central Observatory. The 4 GPS buoys at approximately 4-5 km distance from the RV *Polarstern* (at the map origin). Also shown are the Nloop, Sloop, and Runway transects near the origin. Dash lines show the approximate location of lead and pressure ridge (deformation) areas in October, November-December, January-February, and March-April. The background is the same Radarsat-2 SAR image from 31 December 2019 used in Figure 2.

## 3    Observations used

### 3.1    Initial sea ice thickness and snow accumulation onset

Because the MOSAiC observations started after some sea ice had formed, we defined the initial sea ice thickness and snow accumulation dates based on the bias-corrected reanalysis air temperatures (see Section 3.7) and qualitative information from analysis of the panoramic photography from MOSAiC during October (Itkin et al., 2023). We assumed the following:

1. Predominantly **deformed SYI** was 0.5 m thick on 1 August. Historical summer deployments of ice mass balance buoys show that the ice that survided the summer melt did not start growing before October and that the sea ice thickness in October is a good estimate of the end-of-the-summer sea ice thickness (e.g., Planck et al., 2020). Snow started accumulating on this ice surface when air temperatures were continuously below freezing, assuming that snow accumulation on melting ice is transient. Based on the 10-m air temperature from reanalysis, and assuming the ice surface may have been melting when air temperatures were above -0.1°C, the likely surface freeze-up date was 18 August 2020 (Figure 4). The snow cover appeared thick in the beginning of October (Itkin et al., 2023).





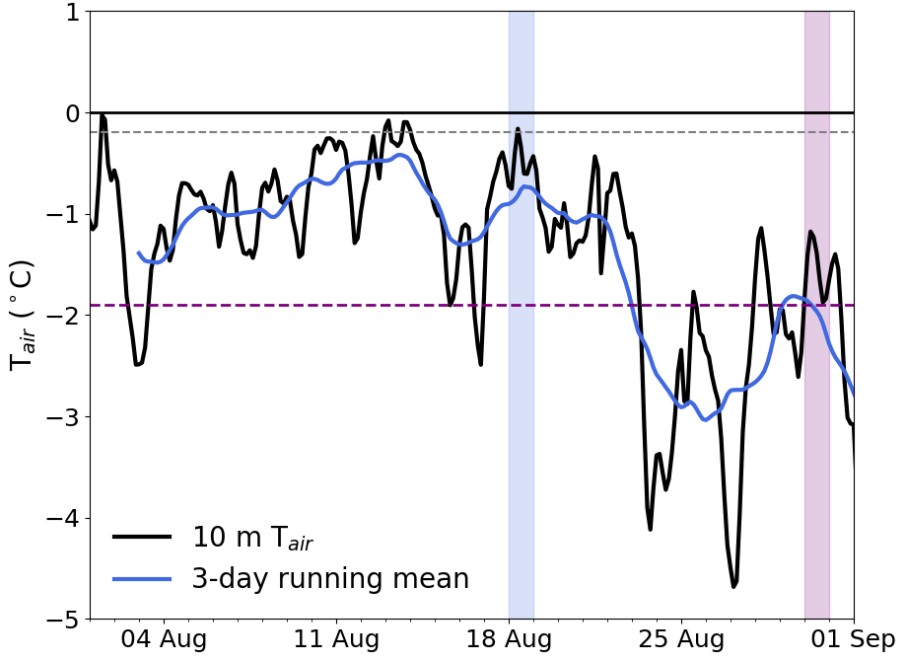

**Figure 4.** 10-m air temperature from 1 August through 1 September 2019 with 3-day running mean. The dates when air temperature and its running mean depart continuously from the freezing temperature of water and sea ice water are identified with blue and purple vertical bars, respectively.

2. Predominantly level and **ponded SYI** was 0.1 m thick on 30 August. After this date the 3-day running mean of air temperatures remained bellow -1.9 °C (Figure 4), the melted-through melt ponds with sea water in them (Macfarlane et al., 2023b) likely began to freeze and snow started to accumulate. The snow cover was thin enough so that dark refrozen pond surfaces were still visible at the beginning of October (Itkin et al., 2023).

3. Predominantly **level FYI** was initiated with 0.05 m thickness on 20 October. The formation of the FYI at the Runway was observed from the panoramic photography (Itkin et al., 2023).

## 3.2 Snow depth

At MOSAiC, snow depth measurements along the transects were collected using an automated snow depth probe Magnaprobe by SnowHydro LLC (Sturm and Holmgren, 2018). The spacing interval of the snow-depth measurements was 1 to 3 m. Because
during ice melt the ice surface is soft and granular, Magnaprobe measurements can erroneously detect melted ice surfaces as snow after melt onset (Webster et al., 2022; Itkin et al., 2023), no Magnaprobe snow depth measurements were used after mid-July. The snow depth measurements at the coring site were read from a graded snow stake. Typically one representative snow depth measurement per ice core was taken and three ice cores were sampled during each approximately bi-weekly



measurement cycle. The snow depth at the stake field was read from installed snow stakes. The stakes were installed in lines
or grids of approximately 5 m distance.

The average level-ice snow depth was calculated as described in the next section.

## 3.3  Ice thickness

The distance from the snow surface to the ice-ocean interface (total snow and ice thickness) was measured using a Geophex Ltd,
GEM-2 broadband electromagnetic induction device (Hunkeler et al., 2015). The estimated precision of such measurements is
approximately 0.1 m. The sea ice thickness was then calculated by subtracting gridded versions of the snow depth and total
snow and ice thickness measurements (Itkin et al., 2023). Sea ice thickness at the coring sites was measured from the core
sampling hole drilled through the ice. Typically three ice cores were extracted. Large deviations in sea ice thickness (larger
than 1 meter from other measurements) were labeled as deformed ice and removed from the analysis. The hot wires in the
stake fields were initially paired with the snow stakes (Raphael et al., 2024), but during the winter many wires were lost.
For all transects, the level sea ice thickness was estimated from the mode of the ice thickness distribution as in Itkin et al.
(2023). To estimate the snow depth on level ice, we averaged the snow depth from all sea ice measurement points where the
sea ice thickness was within 0.01 m of modal sea ice thickness.

## 3.4  Snow density

Snow density was measured in the snow pits by three different methods: 1) by density cutters, 2) by SnowMicropenetrometer
(SMP), and 3) by $SWE$ cylinder. Taylor–LaChapelle density cutters of 3 cm height and 100 m$^3$ volume were used at MOSAiC.
The density was sampled at 3 cm intervals covering the entire snow depth profile. All these cutter measurements were then
used to calculate the bulk snow density. SMP measures force needed to penetrate through the snow depth profile. These force
measurements are directly related to the snow hardness and can be correlated to snow density (King et al., 2020). The SMP
density profiles were also used to calculate the bulk snow density (Wagner et al., 2022). A $SWE$ cylinder with 9 cm diameter
was used to sample a vertical column of snow for total snowpack $SWE$. In addition, snow depth was measured and bulk snow
density was estimated directly from these two measurements.

All three snow density measurements have their advantages and disadvantages. The measurements using density cutters
are the most precise, but also the most labor intensive. Still, 243 bulk snow density measurements from cutters were selected
from the MOSAiC database (Macfarlane et al., 2023b) to represent the three ice types. The advantage of the SMP and $SWE$
cylinders measurements are that they are fast and increase the spatial distribution. In this study we used 182 SMP measurements
sites. Each of these sites was represented by an averaged value for a cluster of five to dozens of distributed measurements at a
given location or transect. This instrument measures snow hardness and snow density estimates are obtained by an empirical
fit (King et al., 2020). It is known (Sturm et al., 2002b), however, that the snow density depends also on the snow texture
(grain size, shape, and bonding) and density estimates of large-grained depth hoar can be overestimated when approximated
using SMP hardness measurements (King et al., 2020). The $SWE$ cylinder bulk snow density measurements are known





to misrepresent the density of a snow cover with large depth hoar crystals, because the depth hoar can collapse during the measurement (López-Moreno et al., 2020). Here we used 166 $SWE$ cylinder measurements.

For the study described herein, the snow density observations described above were used to produce a seamless annual evolution (in a mean, representative in space and time, fashion) of bulk snow density over the MOSAiC Central Observatory. To do this, we fit a straight line through the data ($\rho_s = 0.22x + 253$, where $\rho_s$ is snow density and $x$ is time in units of days since 25 October 2019) to approximate the annual snow cover snow density evolution (Figure 5). As pointed out elsewhere (e.g., Sturm et al., 2002a), the snow densification process is generally slow. The increase of density by approximately 50 kg/m$^3$ between October and May, shown in Figure 5, is relatively small compared to the approximately 100 to 150 kg/m$^3$ density variability found at and between the sampling locations, at any given time. Likewise, the MOSAiC bulk density is seemingly independent of the ice type, as found previously for winter snowpacks (Sturm et al., 2002a; Merkouriadi et al., 2017). In contrast, King et al. (2020) found more depth hoar on topographically variable multi-year ice than on relatively smooth FYI. The MOSAiC situation may be explained by the relatively thin snowpack that is similar for all ice types already by early winter (Itkin et al., 2023).

The seasonal development of bulk snow density episodically deviated from this linear fit. For example, the bulk density is lowest after the snowfall events in December and March-May. While the decrease in December is short and transient, the decrease in March-May is long and prominent in the data. This may be a consequence of a sampling bias towards level ice (Macfarlane et al., 2023b). The logistical challenges connected to sea ice deformation, in that part of MOSAiC, limited the snow density sampling to the measurement locations at the starboard side of the ship, where only level ice sites on SYI were sampled. During spring, Snow 1, Snow 2, and many other locations were not accessible. In addition, the final sampling on Snow 1 provided deeper and denser snow values.

### 3.5 Sea ice deformation

Hourly location data from 4 GPS buoys (2019P103, 2019P193, 2019P195, and 2019P204; Bliss et al. (2023)) within a radius of approximately 5 km from RV *Polarstern* (Figure 3) were used to quantify MOSAiC Central Observatory sea ice deformation. The total sea ice deformation $\epsilon_{TOT}$ (Figure 6d) was calculated from 3-hourly positions (every third hour was extracted from the hourly data) following Hutchings et al. (2012):

$$\epsilon_{TOT} = \sqrt{\epsilon_{DIV}^2 + \epsilon_{SHR}^2}, \tag{1}$$

where $\epsilon_{DIV}$ is divergence and $\epsilon_{SHR}$ is shear calculated by the line integrals. This general sea ice deformation measure was used to avoid the errors associated with deformation calculations that separate the individual contributions from convergence, divergence, shear, and the associated lead and pressure zone processes (Hutchings et al., 2012; Itkin et al., 2017).

These 4 buoys are a subset of the distributed buoy network deployed at MOSAiC (Nicolaus et al., 2022; Bliss et al., 2023). They were selected because their spacing roughly formed a rectangle throughout the entire period from October through May.




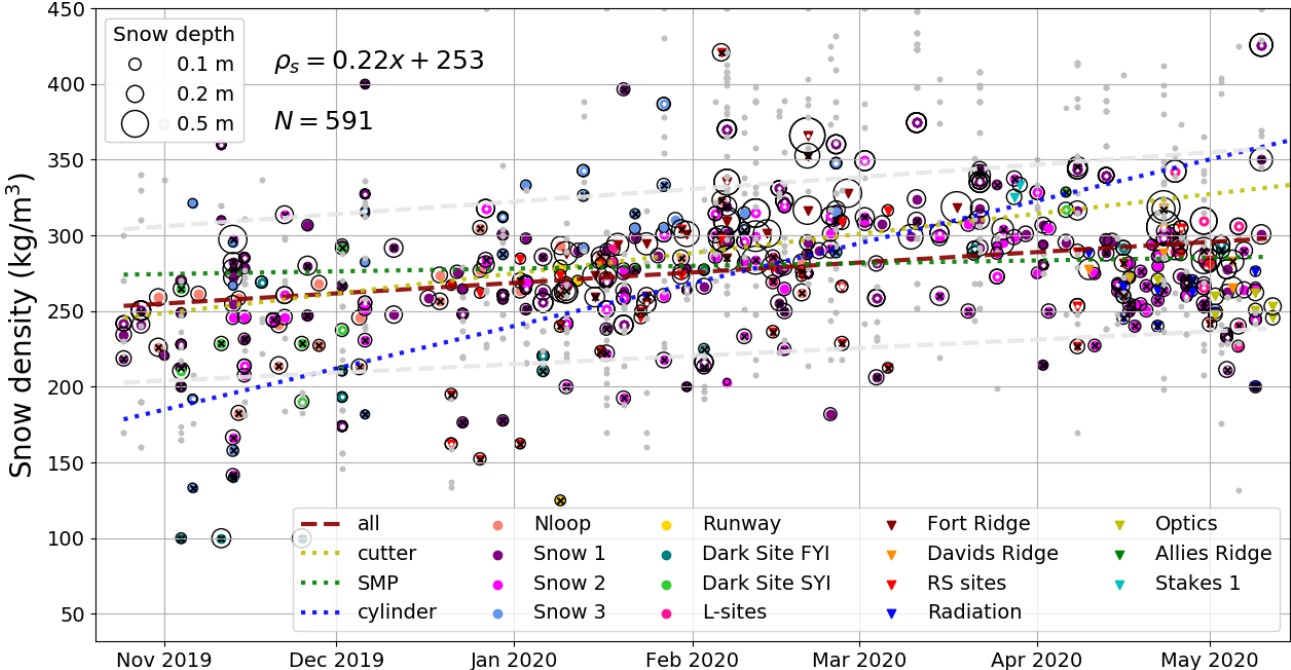

**Figure 5.** Bulk snow density from density cutters, SnowMicropenetrometer (SMP), and $SWE$ cylinder, during the pre-melt period from MOSAiC snow pits. The red dashed line is a linear fit to all the data and grey dashed lines are the limits of lower and upper 20% values typical for depth hoar and wind slab. The yellow, blue, and green lines represent best linear fits for individual measurement subsets. Estimates from the $SWE$ cylinder are identified using a black cross. Estimates from cutters have a white dot. Individual cutter values are displayed as small grey dots.

The coarse resolution sea ice deformation data obtained from the 4 buoys were supplemented by qualitative analyses of relative sea ice motion from the ship radar images (Krumpen et al., 2021a,c,b). The ship radar images cover an area with a radius of 5.4 km around the RV *Polarstern*, and have a spatial resolution of 8 m. These images were collected throughout the MOSAiC expedition every 2 seconds. For this paper, we analysed the temporal changes that occurred in hourly animations of these images to detect the approximate timing and locations of major shear zones and lead openings within the approximate 5 km buoy radius of the RV *Polarstern*.

### 3.6 Ship location data

MOSAiC Central Observatory location data were obtained from autonomous buoy 2019I3 (Bliss et al., 2023). To perform our year-long snow and ice simulation, we required ice parcel location information during the entire simulation year (1 August 2019 through 31 July 2020). To define location coordinates prior to October (first 70 days of simulation), a back-trajectory model (Liston et al., 2018) was implemented and driven with 25-km spatial resolution ice motion vectors from the National Snow and Ice Data Center (Tschudi et al., 2020). We identified the ice parcel the ship was anchored to, and traced it backwards



**Table 1.** Bias correction factors applied (added or multiplied) to MERRA-2 weather data during times when local meteorological observations were not available.

| Air Temperature Threshold (Tair) | Air Temperature Addition (°C) | Relative Humidity Addition (%) | U Wind Speed Component Multiplier | V Wind Speed Component Multiplier | Water Equivalent Precipitation Multiplier |
|---|---|---|---|---|---|
| Tair > 5.0 °C | -0.6 | +4 | 0.91 | 0.88 | 2.13 |
| Tair ≤ 5.0 °C | -2.4 | +2 | 0.95 | 0.86 | 2.13 |

in time to identify the likely ice parcel location history prior to establishing the Observatory. Similarly, ice parcel location data were used to provide ice location data after the buoy malfunctioned in late July (last 7 days of simulation). These procedures resulted in a full-year time series of ice parcel location data (longitude, latitude, and date) that corresponded to the drift of the observatory. As shown on Figure 1 all used drift trajectories coincide well with each other.

## 3.7 Atmospheric data

To drive our model simulations, we used 10-m meteorological observations (air temperature, relative humidity, and wind speed and direction) from the MetCity tower (Figure 2) provided by Cox et al. (2023), and precipitation data from KAZR radar (Matrosov et al., 2022) on board RV *Polarstern*. Any gaps in the observational data were filled using bias-corrected NASA Modern Era Retrospective-Analysis for Research and Applications, Version 2 (MERRA-2; Gelaro et al. (2017)) reanalysis data. Following standard SnowModel-LG procedures (Liston et al., 2020), the ice parcel coordinate data (longitude, latitude, and date) described above were used to identify the nearest MERRA-2 grid cell that the MOSAiC ice parcel corresponded to on each day of the simulation year. This yielded a full year of meteorological forcing data, with no missing values (Figure 6). All of the MetCity, KAZR, and MERRA-2 data were then aggregated to 3-hourly values used in the model assimilations (averages for air temperature, relative humidity, and wind speed and direction, and sums for precipitation).

The biases (offsets for air temperature and relative humidity, and multipliers for the *U* and *V* wind components and precipitation) for each of the atmospheric variables were determined by comparing the averages of the observational and reanalysis data, during time periods when both were available. Separate corrections were made for times when the air temperature was above or below -5.0 °C. Periods with air temperatures above this threshold were approximately 1 August - 15 September 2019, and 15 May - 31 July 2020. The bias corrections are provided in Table 1.

In addition to the basic meteorological variables described above, additional atmospheric forcing data were generated by MicroMet (Liston and Elder, 2006b) and used in our model assimilations, including cloud cover, incoming solar radiation, incoming longwave radiation, and surface pressure.





**Figure 6.** Atmospheric (10 m) and ocean forcing from MOSAiC observations and reanalysis data together with the sea ice deformation data from buoys: a) air temperature and ocean surface heat flux, b) relative humidity, c) precipitation rates and cumulative precipitation, d) total sea ice deformation $\epsilon_{TOT}$ and cumulative $\epsilon_{TOT}$, and e) wind speed and direction. Highlighted by blue, yellow, pink, and green shading are the periods of storms and sea ice deformation that influenced snow accumulation in the MOSAiC observations. The black values in the plots show missing values filled in by the adjusted reanalysis.



## 3.8 Ocean Heat Flux

Ocean surface heat fluxes used in this study loosely follow the estimates of Lei et al. (2022) based on a cluster of 23 buoys
'Sea Ice Mass Balance Array' (SIMBA) designed by Jackson et al. (2013). SIMBAs are equipped with thermistor chains that
measure temperature through air, snow, ice, and ocean; several other researchers have used SIMBA or similar data to estimate
snow depth and sea ice thickness evolution at MOSAiC (Lei et al., 2022; Perovich et al., 2023; Salganik et al., 2023b). In this
paper, only ocean surface heat fluxes from SIMBA were used. The time series were extrapolated by constant values to cover
the period prior to the buoy deployment (Figure 6a). At the end of our analysed period (9-29 July 2020) we adjusted the Lei
et al. (2022) time series to increase from 16 to 44 W m$^{-2}$. This was based on the estimates by Salganik et al. (2023a) using
level ice temperatures and bottom melt rates in the Central Observatory.

# 4   Models used

## 4.1   SnowModel-LG

In this study, SnowModel was used to represent snow-on-sea-ice processes and evolution. SnowModel has its origins as a
terrestrial, multilayer, spatially distributed, snow evolution modeling tool (see Liston and Elder (2006a), and the references
contained therein). It is coupled to a high-resolution atmospheric model called MicroMet (Liston and Elder, 2006b) that pro-
vided surface forcing (based on the 10-m meteorological forcing summarized in Figure 6) to SnowModel (Liston and Elder,
2006b), and SnowAssim that assimilates available field and remote sensing observations (Liston and Hiemstra, 2008). Adap-
tation of these models to Lagrangian drifting sea ice environments and ice parcels (SnowModel-LG) was described by Liston
et al. (2018) and Liston et al. (2020). In this study, we used a 1-D (vertical profile) version of the multi-layer modeling system to
create high resolution time series (3-hour time increment) of MOSAiC snowpack development and evolution. In SnowModel,
$SWE$ is calculated by solving the snow source and sink equation:

$$\frac{dSWE}{dt} = \frac{1}{\rho_w}[(P_R + P_S) - (S_{SS} + S_{BS} + M) \pm D], \tag{2}$$

where $SWE$ is snow water equivalent (m); $\rho_W$ is water density (1,000 kg m$^{-3}$); $P_R$ and $P_S$ represent sources of snow
cover from rainfall and snowfall; $S_{SS}$, $S_{BS}$, and $S_M$ represent the snow cover sinks through static sublimation, blowing snow
sublimation, and melt; and $dt$ ($s$) is the model time increment (= 10,800 $s$ = 3 hours in this application). Finally, $D$ represents
the sea ice deformation snow sink or source. The units of all sources and sinks is kg m$^{-2}$ s$^{-1}$. Snow depth $h_s$ (m) is estimated
as:

$$h_s = SWE\frac{\rho_w}{\rho_s}, \tag{3}$$

where $\rho_s$ (kg m$^{-3}$) is snow density.





## 4.2 HIGHTSI

In this study, a 1-D thermodynamic sea ice ice model called HIGHTSI (Launiainen and Cheng, 1998; Cheng et al., 2008; Merkouriadi et al., 2020) was used to simulate sea ice thickness evolution. In this application, the snowpack used in HIGHTSI was provided by our SnowModel-LG assimilations. In HIGHTSI, the snow thermal conductivity, $k_s$, was parametrized using the snow density, $\rho_s$, based on a second-order polynomial fit from MOSAiC snow pit data following Macfarlane et al. (2023a):

$$k_s = 2.62e^{-6}\rho_s^2 + 1.54e^{-33}\rho_s + 3.04e^{-2} \tag{4}$$

This parametrization results in a seasonal bulk snow thermal conductivity increase from $0.21\ WK^{-1}m^{-1}$ in October 2019 to $0.27\ WK^{-1}m^{-1}$ in May 2020. Its values coincide well with estimates of Raphael et al. (2024), who provided an insightful discussion of the quality of the snow thermal conductivity estimates at MOSAiC. The ice thermal conductivity was constant in HIGHTSI throughout the winter and set to $2.03\ WK^{-1}m^{-1}$. During the melt period, ice conductivity depends on the surface temperature and salinity. For our HIGHTSI simulations, during melt, it was set to vary between 1.7 and $2.1\ WK^{-1}m^{-1}$, 295 following the ice surface conductivity measurements made during MOSAiC (Macfarlane et al., 2023a).

## 5 Simulations and results

To satisfy our general goal of using a collection of modeling tools (SnowModel-LG and HIGHTSI) to fill in temporal gaps in MOSAiC field observations, we made the following assumptions and performed the following tasks. First, we assumed the MOSAiC field observations were perfect; we made no effort to correct them for any potential measurement errors, others have 300 taken on those tasks (e.g., Cox et al., 2023; Matrosov et al., 2022; Itkin et al., 2023). In addition, we configured our modeling tools to fill data gaps between those observations with modeled values that were as realistic as possible. Our general vision was to produce snow and ice time-series information, with no missing data, covering a full snow and ice evolution year, at a 3-hour time increment, that contained MOSAiC snow and ice observations when they existed, and physically realistic values during any and all periods when they did not. 305 The procedures described below were implemented over the three ice types described in Section 2.1.

### 5.1 Snow calculations

Our snow simulations began by using SnowModel-LG to solve Equation 1, with the ice dynamics term, $D$, set to zero, using the atmospheric forcing data described above. Figure 7a presents a comparison of the $SWE$ observations and the simulated $SWE$ time series for the deformed SYI ice type (Nloop transect); the other two ice types are similar and not shown. 310 To calculate the snow depth evolution, the final $SWE$ evolution defined above was further modified using our MOSAiC snow density observations. The density equation associated with the line fit in Figure 5 was used to define the observed snow density at the $SWE$ observation times. We then calculated the ratio of this observed snow density to the SnowModel-LG produced snow density at those times. This created a density correction parameter that, when multiplied by the model simulated



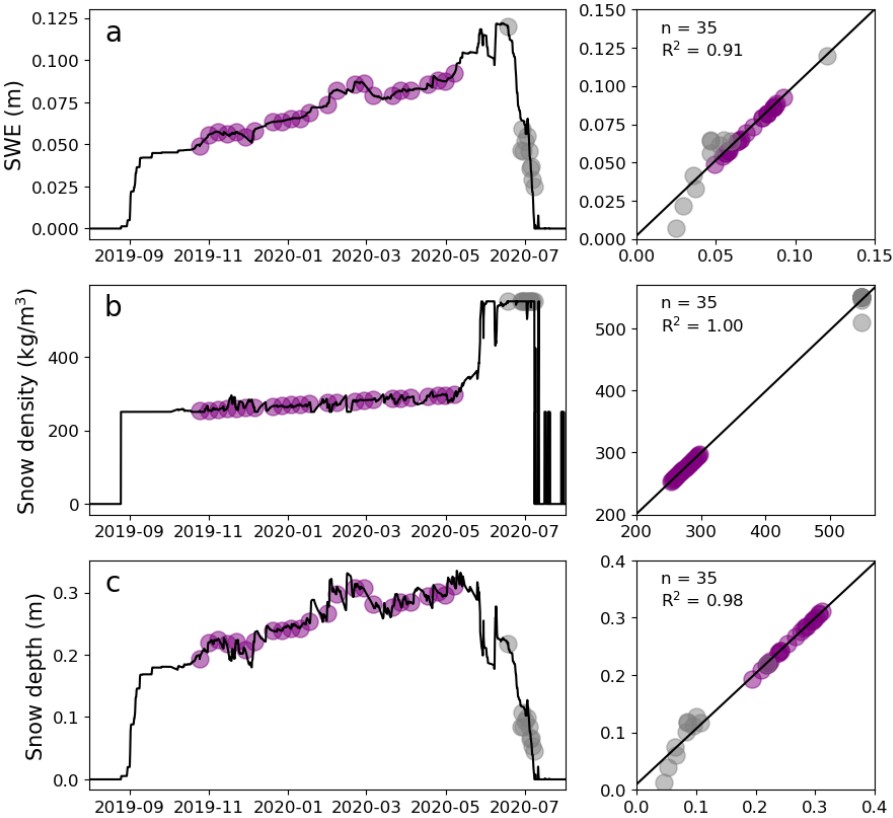

**Figure 7.** Nloop SnowModel-LG snow assimilation time series and scatter plots (x-axis for observations and y-axis for model) for a) SWE, b) bulk snow density, and c) snow depth in each of the figure rows. Assimilated values are represented by purple circles. Melt period values (represented by grey circles) were not assimilated.

snow density, reproduced the observed density. Similar to our $SWE$ adjustments, these correction parameters were linearly
interpolated between the observations, with the correction set to unity before the first observation and after the last observation. This correction time series was then multiplied by the model simulated snow density to create a temporally continuous density evolution that matched the density observations when they occurred, and filled in realistic density values during periods with no density observations. Figure 7b presents a comparison of the snow density observations and the simulated density time series for the deformed SYI ice type (Nloop transect); the other two ice types are similar and not shown.

Snow depths over the annual simulation period, with 3-hourly time increment, were then created using Equation 2, with inputs of the 3-hourly $SWE$ and snow density data described above (Figure 7c).





## 5.2 Sea ice deformation snow sink and source calculations

To understand the role of ice deformation, $D$, in the evolution of snow properties, the difference between modeled and observed $SWE$ was assumed to equal $D$ in Equation 1 (model minus observed; so positive $D$ values represent a $SWE$ sink, or a $SWE$ loss resulting from ice dynamics). Then, to create a continuous $D$ time series at the 3-hour time increment, $D$ was linearly interpolated between the sea ice freeze-up date (zero snow depth) and individual $SWE$ observation times. Before and after these dates $D$ was set to zero. This $D$ term was then subtracted from the model simulated $SWE$ to create a temporally continuous $SWE$ evolution that matched the $SWE$ observations when they occurred, and filled in realistic $SWE$ values during periods of no $SWE$ observations. Figure 8 presents the resulting time evolution of $D$ and $SWE$ for the three types of MOSAiC ice.

## 5.3 Thermodynamic ice growth calculations

The temporally continuous snow depth and density evolution over the three ice types, the atmospheric forcing, and the ice-ocean interface heat fluxes, were used to force the HIGHTSI thermodynamic sea ice model. The snow simulated in the previous section represents the mean snow cover, including both snow on level sea ice and snow on deformed ice. Because the modal sea ice thickness only represents level sea ice thickness, and since HIGHTSI only simulates thermodynamical sea ice growth, we performed separate SnowModel-LG simulations over each of the three ice types, assuming level-ice only, to create the snow forcing for the HIGHTSI sea ice simulations. This was done by repeating the calculations described in the previous sections, while only using $SWE$ observations collected over level, undeformed ice.

In addition, to account for the role of meters-scale snow depth variability resulting from snow bedform (e.g., snow dunes, sastrugi, etc.) snow-depth variability found on level ice, one standard deviation of snow depth was removed from the simulated level-ice snow. This step effectively accounts for the role that thin snow, in the troughs between the snow bedforms on level ice, has on enhanced ice growth (Sturm et al., 2002b; Liston et al., 2018), and can be thought of as a simple, bedform-scale heat transfer parameterization.

Approximately 16% of all level-ice snow depth measurements along the transects had snow depths at or lower than the mean level ice snow depth minus one standard deviation. This corresponds to the proportion of troughs between the snow bedforms in the MOSAiC snow transect observations. Our simulations used observed MOSAiC snow depths, and observed snow and ice thermal conductivity (Macfarlane et al., 2023a), to drive our ice growth simulations. Figure 9 presents the comparison of the ice thickness model simulations and observations for the three types of MOSAiC ice. For each ice type, model simulations with mean snow depth (level and deformed ice), level ice snow depth, and level ice reduced by one standard deviation, are presented.





**Figure 8.** Transect observations and SnowModel-LG simulations of $SWE$ for a) Nloop, b) Sloop, and c) Runway. Input precipitation from snow accumulation onset and time derivatives of $D$ ($\frac{dD}{dt}$) and $\epsilon_{TOT}$ ($\frac{d\epsilon_{TOT}}{dt}$) are also displayed. Highlighted by blue, yellow, pink, and green shading are examples of periods of storms and sea ice deformation that influenced the ice-type specific snow accumulations. Signs of dD/dt discriminate between source and sink.





**Figure 9.** Seasonal development of snow and ice cover from model and observations for a) Nloop, b) Sloop, and c) Runway. The error bars of observations represent one standard deviation. The grey shading around the sea ice thickness modes estimates indicates the 0.1 m precision of the GEM-2 method. The snow depth scale is exaggerated four-times compared to sea ice thickness.




## 6 Discussion

As demonstrated in over 200 publications, MicroMet, SnowModel, SnowModel-LG, and SnowAssim, have been shown to reproduce a wide range of snow-related observations found in terrestrial and sea ice environments around the world (e.g., see the references contained within Liston et al. (2020) as a representative subset of these publications). This study, and the analyses presented herein, provides four opportunities that are unique to those other studies: 1) to identify snow and ice processes and evolution that likely took place during the MOSAiC snow and ice evolution year before the ship arrived, and during winter, spring, and summer periods when the ship was not stationed in the ice or snow and ice measurements were not possible for some reason; 2) to help understand and quantify the role of ice dynamics on snow mass budgets; 3) to help understand and quantify the role of the resulting snow distributions on sea ice growth and decay; and 4) to help understand the processes of the early melt season, including estimating timing of melt onset of snow, first snow-free date, and melt onset of ice. The following discussion analyzes these four issues.

### 6.1 Missing time periods during the MOSAiC observation year

Using an atmospheric reanalysis that was bias corrected by MOSAiC observations, atmospheric observations from the MO-SAiC observatory, and extrapolated ocean heat flux estimates, we used SnowModel-LG with assimilated snow observations, and HIGHTSI, to create a seamless snow and ice property time series with a 3-hourly time step for the entire ice year from 1 August 2019 through 31 July 2020. The model simulations were critical to fill in three kinds of missing data: 1) the time period before the ship's arrival and the start of observations, 2) missing data due to weather and logistical reasons, and 3) times between the regular discrete observations of snow properties and sea ice thickness.

Before the ship's arrival, we used bias-corrected atmospheric reanalysis data to estimate the snow accumulation and freeze-up time for the three ice types (see Section 3.1). Although the timing of freeze-up is critical for the snow and ice cover, this period is logistically challenging for any ship-based snow and ice observation expeditions (Nicolaus et al., 2022) or autonomous instrument programs (Benjamin Rabe, 2024).

Occasionally, during winter, the continuous weather data collection or weekly snow and ice sampling was interrupted due to ice break-up, weather, or logistical reasons, such as crew exchanges (Cox et al., 2023; Matrosov et al., 2022; Itkin et al., 2023; Nicolaus et al., 2022). The bias-corrected atmospheric reanalyses is of sufficient quality that SnowModel-LG (apparently) accurately reproduced these missing periods. Similarly, there were no atmosphere, sea ice, and snow observations collected between 7 May and 15 June. The last snow measurement assimilated into SnowMolde-LG was on 7 May (Figure 7). Still, the model accurately reproduced the first observations collected in June, more than than a month later. SnowModel-LG also simulated the transient melt onset at the end of May, when first extensive melt ponds of the season were widely observable on satellite images (Webster et al., 2022). This means that, by fusion of observational data and numerical models, we successfully bridged the periods when no spatially distributed data were collected at MOSAiC.

The 7-day observation period of the MOSAiC transects was chosen to resolve the synoptic cycle of snow accumulation and ice growth. The data-model fusion we implemented here has a full year time series (1 August 2019 through 31 July 2020), at a





3-hour time increment that resolves the timing of any atmospheric events with a high temporal precision. This is of relevance
for understanding snow and ice processes as well as atmosphere - snow - ice - ocean interactions. In addition to the suite of
snow and ice related variables presented herein, these simulations come with numerous other surface energy flux and mass
balance variables that are all internally consistent with each other; this is a numerical requirement of all the modeling tools
used in this study.

## 6.2    Sea ice deformation can be an important snow source or sink

Previously, Liston et al. (2020) demonstrated that sea ice deformation, calculated as a residual using coarse resolution atmo-
spheric and ice concentration and movement forcing data at the pan-Arctic scale, was highly correlated to the sea ice drift and
the new ice formation associated with it. In this study, we extend their findings to a local scale by examining the connection
between snow mass balance and sea ice deformation (herein formulated as $D$, see Equation 2). Using observed and estimated
atmospheric forcing data, and periodic $SWE$ and snow density observations, SnowModel-LG reproduced the observed snow
evolution on the three sea ice types with different age and sea ice deformation characteristics found at MOSAiC (Figure 8).

On all three ice types, the strongest winter season sinks were static and blowing snow sublimation ($S_{SS}$ and $S_{BS}$) which by
7 May cumulatively removed 68, 68, and 70 % of $SWE$ from snowfall ($P_S$) in Nloop, Sloop, and Runway, respectively. This
is represented by the difference between 'precipitation' and 'model no D' on Figure 8. Note that, in this environment, if the
blowing snow is not captured by a ice-topographic drift trap, or blown into an open lead, it blows perpetually and, in air that
has a humidity deficit, it eventually sublimates completely away (Tabler, 1975; Liston and Sturm, 2004). These $S_{SS}$ and $S_{BS}$
values were about three times as large as in Liston et al. (2020). This is likely due to the specific weather during MOSAiC
winter and location during the drift, including, generally low snowfall ($P_S$) after freeze-up, frequent storms with high winds
(Rinke et al., 2021), and relatively high sea ice concentration (Krumpen et al., 2021) with low near-surface relative humidity
during winter. $P_S$, $S_{SS}$, and $S_{BS}$ operate at synoptic temporal and length scales, and were the same (or very similar for $S_{SS}$
and $S_{BS}$, which depend on grain bounding) for all ice types.

The differences in $SWE$ evolution on the three ice types were largely controlled by the ice (and snow) onset date, and the
differences in the remaining wintertime snow sink or source - the ice dynamics term $D$. This is represented by the difference
between 'model no D' and 'model with D' on Figure 8. $D$ is the only simulated local source or sink; in the natural system,
$D$ produces ice roughness features such as rubble ice and pressure ridges, and lead timing, size, and distribution. Following
any sea ice deformation, a certain amount of airborne snow will be removed to open water in the leads (Clemens-Sewall et al.,
2022) or stored in snowdrifts at the deformed ice roughness features (Liston et al., 2018; Itkin et al., 2023). During winter, the
wind velocity is frequently above the blowing threshold value (7.7 m s$^{-1}$) following Li and Pomeroy (1997), which provides
the justification for our parametrization of $D$ as a sea ice deformation snow sink or source (see Section 5.1). After melt onset,
the snow grains are wet and no drifting snow is observed (*sensu* Pomeroy et al., 1997). Following this principle, $D$ was set to
zero in May after the last transect measurements.

$D$ remained small throughout the simulations, but its accumulated affect by 7 May (the last winter observation) was 10, 8,
and >1 % in Nloop, Sloop, and Runway, respectively. $D$ is likely large right after freeze-up; this is a period of thin ice with



lots of deformation. More studies of this fast-changing period with thin ice are needed to understand what exactly is happening

when the ice first forms. The importance of erosion for $SWE$ at MOSAiC was explored by Wagner et al. (2022), who gave

estimates of erosion based on uncalibrated snowfall rates, wind speeds, and $SWE$ in Sloop and Nloop. While the magnitude

of the combined snow sink by Wagner et al. (2022) is similar to ours (53-68%), their study could not differentiate between

erosion and sublimation. Our study shows that $D$ was predominantly a sink in the case of Nloop and Runway and, after ridge

formation in November, was occasionally a source in Sloop.

Strong winds are generally associated with synoptic events that often cause sea ice deformation and bring precipitation. For

Sloop, high wind speeds had no significant statistical correlation with the time derivative of $D$, but we found a strong ($R^2$=67%,

N=16, p=0.0001) linear correlation between the absolute value of this derivative to the derivative in accumulated total sea ice

deformation - $\epsilon_{TOT}$ estimated from the GPS buoys (see Section 3.5). For Nloop, using the same sampling steps in time as

for Sloop, the correlation was similarly strong ($R^2$=62%, N=16, p=0.0016). For Runway, the sample size was too small (N=2)

to perform this analysis, but a combined analysis for all locations shows a strong correlation too ($R^2$=59%, N=33, p=0.0001,

Figure 10)). Note that we used the absolute values of the $D$ derivative, since $\epsilon_{TOT}$ is also spatially averaged and differentiation

between sink and source is not possible with these data.

Our findings confirm that the precipitation observations have sufficiently low errors that a minor signal such as snow redistri-

bution due to sea ice deformation was able to be detected. This was, in turn, also only possible because the *in situ* transect and

sea ice deformation observations were frequent enough to resolve the synoptic cycle's contributions to snow and ice evolution.

To understand the reasons for these statistical correlations, we tracked the timing and location of major leads and pressure

ridges (Figure 3) occurring in the vicinity of the MOSAiC transects on the ship radar images (see Section 3.5). Such qualitative

analyses allowed us to distinguish between the sink and source direction of $D$. We found that any close upwind deformation

was associated with a $SWE$ decrease. Any deformation inside the transect, or just adjacent to it, was connected to a $SWE$

increase that can not be explained solely by snowfall. For example, in late October and November, pressure ridges formed

north of Nloop (blue line on Figure 3). This was accompanied by SE winds that accumulated snow in Nloop - just some 100

m upwind of the new ridges. In November and December a new lead and ridge line were created between Sloop and Nloop

(yellow line on Figure 3). This time, the winds were strongly SW. Nloop was upwind and a new ridge formed inside Sloop.

Sloop gained $SWE$ and Nloop lost it. At the end of January and beginning of February a lead was active east of Sloop (pink

line on Figure 3). The W winds first caused a local source of snow in Sloop. When winds turned to E, the lead was the local sink

that caught snow and prevented deposition in Sloop. Nloop was too far away to be affected. In March and April, a large lead

separated Nloop and Sloop once again (green line on Figure 3). The winds were mainly S, along the lead, but with occasional

W winds. The upwind Sloop lost a considerable amount of snow. Because the winds were E in May, Sloop again gained some

snow during this time. Nloop was further away from the deformation zone, and mainly lost snow during this period too.

The strong correlation between derivatives of $D$ and $\epsilon_{TOT}$, for these major deformation events, strongly suggests that there

was an immediate response to the creation of new roughness elements and open water areas. This is in contrast to the slow

accumulation occurring in the old roughness features created during previous deformation events. Spatially distributed, 3-

dimensional, simulations, similar to those performed by Liston et al. (2018), over a dynamic topography that quantifies sea ice





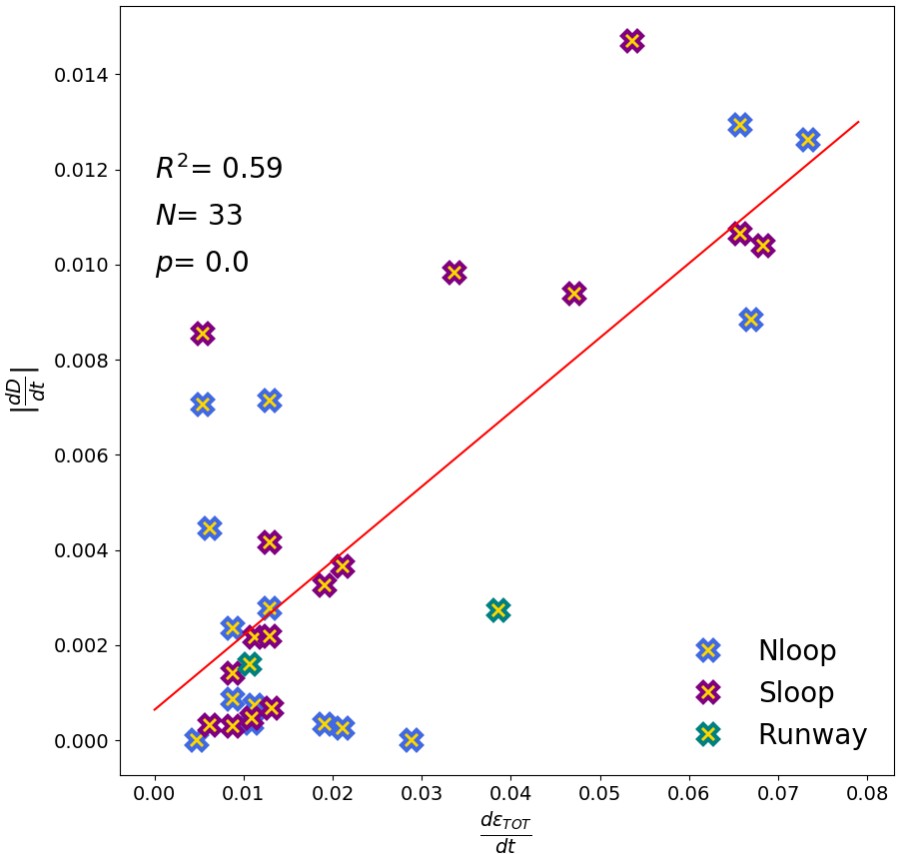

**Figure 10.** Scatter plot of derivatives of $D$ and the cumulative total deformation. The time steps between observations in Nloop and Sloop are the same (some observations during quiescent periods are skipped in the Nloop that had more frequent observations than Sloop).

deformation at much higher resolution than a handful of GPS buoys, are necessary to estimate how much snow is stored on new rough ice, how much is lost to open water, and how much sublimates away. Such a study would also be able to confirm

or reject a recent observational study, based on spatially very constrained MOSAiC ice core data in refrozen leads (N=5), that indicated very little snow was lost to leads (Clemens-Sewall et al., 2022). Liston et al. (2020) presented a list of eight reasons why snow blowing into leads is likely a minimal component of the snow-on-sea-ice moisture budget. In contrast, loss of snow into leads is used as a tuning parameter in some climate model simulations (Petty et al., 2018; Schröder et al., 2019).

The large magnitude of $D$ in Nloop may be a peculiarity of MOSAiC. The MOSAiC snowpack was thinner than the clima-

tological mean (Itkin et al., 2023), and snowdrifts in pressure ridges and deformed ice can store all snow volume if there is very little snow. This is a known phenomena well researched in terrestrial systems (e.g., Tabler, 1975; Benson and Sturm, 1993; Sturm et al., 2001; Liston et al., 2016, 2024). In windy environments with little snow, all snow will be trapped in the ridges or other ice-surface roughness features. Over the course of the winter, the fraction of snow volume in topographic drift traps





should decrease as the trap fills to capacity and the snow depth in other (non-snowdrift-trapping) areas increase. At MOSAiC,
this never happened, because there was very little snow and newly deformed ice was frequently created. As evidence of this,
the snow depth standard deviation in the pressure ridges increased throughout the winter, because they were never completely
filled to their maximum snow-holding capacity (Itkin et al., 2023).

The $SWE$ on all ice types was different at the start of the MOSAiC observations, but it became increasingly similar towards
the end of the snow accumulation season. Since all three ice types were adjacent to each other and part of the same turbulent
wind field, air temperature conditions, and precipitation forcing, some snow appears to be transported from the oldest and least
dynamic ice type (Nloop), to the most dynamic (Sloop) and youngest (Runway) ice types. This is another hypothesis that could
be explored by high-resolution, spatially distributed simulations that cover various sea ice types with different ice-roughness
characteristics. Such a study would benefit from some kind of snow particle transport accounting within the modeling system,
that would identify and quantify the origins and deposition locations of snow particles being redistributed by the wind.

### 6.3 Impact of snow on sea ice growth

As already shown by Itkin et al. (2023) and Raphael et al. (2024), younger and thinner sea ice types at MOSAiC with initially
thinner snow (Sloop and Runway), had faster sea ice growth rates early in the growth season than the oldest sea ice type
with initially deeper snow (Nloop). Similar findings have been demonstrated in previous observational (Sturm et al., 2002a;
Provost et al., 2017; Rösel et al., 2018) and modeling studies (e.g., Notz, 2009). At MOSAiC, the relatively low initial sea ice
thickness of the SYI (Section 2.1), and the differences in growth rates, led to younger and thinner sea ice thicknesses that were
approximately equal to that of the oldest and thickness ice type by as early as mid-November 2019 (Figure 9).

The importance of the snow accumulation onset for ice growth and spring sea ice thickness is visible even for various ages
of FYI (Figure 9c). There were two ages of FYI that were sampled at MOSAiC. The FYI at the coring site was formed at
freeze-up, on about 1 September, and accumulated practically the same amount of snow as Sloop. This is also the FYI onset
date used by von Albedyll et al. (2022) in a large-scale MOSAiC study. The Runway and Fort Ridge FYI formed about a month
later in leads, and they missed the snowfall during that period. The Ridge Ranch stakes at the Fort Ridge were close to a ridge
and accumulated snow the fastest. The FYI with thinnest snow grew the thickest by the end of winter. This is visible in the
observations (May) and the model simulation, and suggests that the variability of the level FYI thickness can be larger than the
variability of the level SYI thickness.

The high correlation ($R^2$=0.94,N=44) of simulated and observed winter sea ice thickness (Figure 11) is clear evidence of the
importance of the local snow depth and density for the ice growth. Besides these two local variables simulated by SnowModel-
LG, HIGHTSI was also forced by atmospheric temperature (Section 3.7) and ocean heat fluxes (Section 3.8) that were assumed
uniform for all MOSAiC ice types. In addition, two storms increased the snow depth variability at MOSAiC: 1) the November
storm (yellow shading in Figures 6 and 8) increased the standard variability in Nloop and Sloop, and 2) the January-February
storm (pink shading in Figures 6 and 8) increased the standard variability on Runway. Both events were followed by an increase
in level-ice thickness. This indicates that the developed snow bedforms promote loss of heat from the ocean, and highlights the




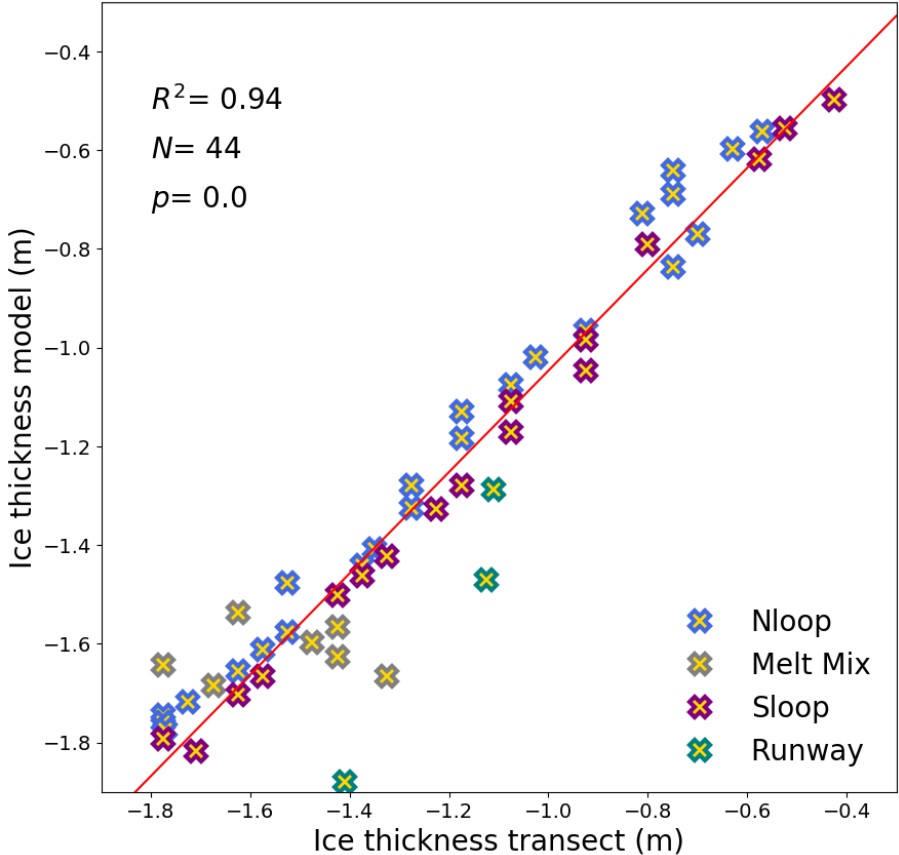

**Figure 11.** Scatter plot of modal sea ice thickness from transect observations and simulated sea ice thickness. The linear correlation coefficients are estimated just for the winter transects (Nloop, Sloop, and Runway).

relative importance that local-scale (locally one-dimensional or fully three-dimensional) heat fluxes, though the snow and sea ice, play in governing ice growth (as suggested by Sturm et al. (2002b), Liston et al. (2018), and Itkin et al. (2023)).

    Our bedform parametrization, as described in Section 5.3, is an example of how these local-scale, sub-grid phenomena and

processes can be accounted for in climate system models in order to accurately reproduce observed sea ice growth. Instead, current climate models often use thermal conductivity values that are approximately 50% larger that those measured (Macfarlane et al., 2023a; Raphael et al., 2024) and used here (Section 4.2); effectively increasing the heat transfer through the snow in an effort to reproduce observed ice growth. The results presented herein suggest that climate system models need to be accounting for local-scale heat transfers, instead of adjusting the snow thermal conductivity away from observed values. Sturm

et al. (2002b) also came to this conclusion over two decades ago.



The ocean surface heat fluxes during winter were low (Figure 6a) and the ocean surface was very cold, or even supercooled Katlein et al. (2020). During this time, the ice growth was mainly governed by the low atmospheric temperatures and the snow thermal conductivity.

Our estimates of initial thickness (Section 2.1) and snow accumulation onset dates (Section 3.1) are unfortunately crude.
More observations from that period, in addition to distributed model runs, may provide better insights into snow and ice initiation and evolution during that period.

### 6.4   Snow and ice melt onset

Snow melt began while *RV Polarstern* was not present in the MOSAiC Central observatory, during the time window between 10 May and 15 June (Nicolaus et al., 2022). Several authors have attempted to estimate this date based on point measurements
from buoys with thermistor chains (Lei et al., 2022; Perovich et al., 2023; Salganik et al., 2023b; Raphael et al., 2024) or satellite imagery (Webster et al., 2022). In our simulations, the snow depth starts gradually decreasing simultaneously with the snow density increase shortly after the last transect measurement on 7 May (Figure 7). This coincided with a storm with strong winds and air temperatures near zero (Figure 6a) and a snow depth decrease detected by thermistor chains (Salganik et al., 2023b). Other thermistor-chain-based studies estimated snow melt onset at much later dates: on 15 May (Lei et al., 2022)
and 8 June (Perovich et al., 2023). The air temperatures were positive for the first time with the storm on May 27 (Figure 6a), coinciding with rainfall (Figure 6c) and a decrease of both $SWE$ and snow depth (Figure 7). The density increased abruptly on 27 May and reached its maximum simulation value (550 kg/m$^3$) on 28 May. At this point in the model, the snow was isothermal and saturated with water. Melt ponds were detected by the thermistor chains as early as 27 May (Salganik et al., 2023b) and appear fully developed on satellite images starting on 28 May (Webster et al., 2022).

The second phase of the end-of-May storm brought cooler air and snowfall (Figure 6a and c) that stopped the melt. $SWE$ recovered and even increased to its maximum value between 7 and 17 June. However, snow density remained at its maximum value and the snow depth continued to decrease despite transient increases with snowfalls at the end of May and in early June (Figure 7). Simultaneously with the $SWE$ winter maximum in early June, snow depth values were similar to early winter values. Both $SWE$ and snow depth fit well with the Melt Mix transect measurements on 17 June - the first snow measurements
after the return of the *RV Polarstern* (Figures 7 and 8). None of the Melt Mix transect measurements were assimilated in our model run. The second and very abrupt decrease in $SWE$ and snow depth started immediately after 17 June. All snow melted by 8 July, which fits well with the transect observations (Webster et al., 2022; Itkin et al., 2023). Figure 9 shows that, on level ice, the snow was melted completely about 1 week earlier. On level ice with snow reduced by one standard deviation, snow was fully melted even earlier - about 3 weeks prior to the average snow cover. This coincides well with the estimates from eight
thermistor chains deployed on level ice (Lei et al., 2022).

According to transect and point observations on Figure 9, sea ice growth rates dropped towards zero in mid-April. At the same time, several previous MOSAiC studies indicated melt of deformed ice (Raphael et al., 2024; Salganik et al., 2023b; Itkin et al., 2023). The level ice thickness in June however, exceeded April/May values and showed that the ice was growing again in the cold weather and snow depth maximum of late-May and early-June. This is confirmed by our simulations that coincide



with the measurements, despite them not being assimilated. Sustained ice melt started at the end of June, coinciding with the snow-free date on level ice and preceding the abrupt increase in ocean surface heat fluxes by about a week (see Section 3.8). This also coincides with the analysis of the transect (Webster et al., 2022; Itkin et al., 2023) and ice thickness measurements from stakes and coring (Figure 9). Simulated ice-melt lagged behind the thermistor chain data estimates by a couple of weeks (Lei et al., 2022; Perovich et al., 2023), potentially indicating preferential surface melt in the melt ponds and surrounding

highly conductive thermistor chains.

## 7 Conclusions

Virtually any field campaign may include time periods of interest when observations were unable to be made. The MOSAiC field expedition was no exception in this regard. In particular, the MOSAiC sea ice environment included logistical and safety considerations associated with weather, thin ice, and late freeze-up. These often made direct sampling difficult, and contributed

to observation data gaps. In addition, as the Arctic continues to warm, difficulties measuring snow and ice will likely continue. Our data-model fusion methodology presents a mechanism by which those data gaps can be filled. Here, we have combined physics-based modeling tools, with temporally incomplete measurements, to create a full annual time series of 3-hourly snow and ice property values that match the observations when and where they occurred. Finally, the time series data contain realistic values when observations were not available.

Our data-model fusion methodology applied to MOSAiC indicated that:

- The freeze-up date and initial ice thickness estimated in Section 2.1 are physically realistic. The freeze-up dates for the deformed SYI, ponded SYI and FYI were on 18 August, 30 August, and 20 October, respectively. Their initial ice thicknesses were close to 0.5 m, 0.1 m, and 0.0 m, respectively.

- The sea ice thickness of level ice on all three ice types became similar already in December, and reached 1.8-1.9 m in
mid-April. Afterwards, the sea ice growth stopped and then plateaued until the melt onset during the last week of June. The maximum sea ice thickness was 1.9 to 2.0 m for all ice types.

- The snow depth on level ice on all three ice types reached its maximum value of between 0.33 to 0.35 m in the first half of May. Afterwards it was decreasing due to the wetting of the snow. At the same time, the maximum SWE values were reached in mid-June.

- The snow-free date was reached simultaneously on all ice types, after a very rapid melt during the second half of June. Level ice areas became snow-free during the last week of July.

- The sea ice started to melt first from the top surface, with ice-melt onset coinciding with the snow-free date.

The correct initialization of our simulations proved to be the most critical aspect of our work. In this study, we defined the ice and snow initial conditions by analyzing atmospheric reanalysis data. Future snow and ice evolution studies, similar to

MOSAiC, would benefit from actual measurements of the conditions during the freeze-up period. These include challenging



conditions like open water and thin ice; such measurements are not easy to make, but would lend key insights into snow and ice formation and evolution during this critical period; a period that we know very little about.

In addition, this work identified two climate-relevant processes that operate at relatively fine spatial scales. Here we summarize them and suggest ways to use them in other climate-system applications:

– Sea ice deformation was identified as a significant snow trap or sink (Clemens-Sewall et al., 2022; Itkin et al., 2023). High quality precipitation data collected at MOSAiC led to simulations with a relatively small residual, or difference between the simulated and observed snow depth. We found that this residual term can, in large fraction (50-60%), be explained, in a statistical sense, by the regional scale deformation that occurred over the broader area where the deformation observations were collected. In the parametrization developed here (see Section 5.1), we add or remove the

amount of snow in a way that the simulations and observations are numerically complementary. Over bigger domains, the large-scale deformation simulated by regional sea ice models could be used to determine the magnitude of this term.

    – Snow bedform patchiness was identified as a key control influencing area-averaged heat fluxes through the ice and ice growth (*sensu* Sturm et al., 2002b). Local ice growth is likely a result of heat fluxes over several-meters scale footprints; these footprints have diverse snow depths, where the shallowest snow depth has the highest impact on ice growth. In the

parametrization developed here (see Section 5.3), we use the snow depth reduced by one standard deviation to achieve the heat fluxes required to produce realistic sea ice thickness.

Further testing of both parametrizations mentioned above is planned using a high spatial resolution snow and sea ice model, forced by high resolution sea ice deformation data based on, e.g., quantitative analyses of relative sea ice motion from the ship radar images similar to Oikkonen et al. (2017).

Our model configurations can be considered single column simulations, but each single column represents approximately 1500 point measurements designed to be representative of a much wider area. This work accounted for the three most typical sea ice types found in the MOSAiC Central Observatory; these three ice types also represent the three ice types most commonly found throughout the modern Arctic. Combining the results and procedures presented herein, with the knowledge of their Arctic-wide spatial distribution, would allow similar analyses to be performed and used to estimate the annual evolution of the

Arctic heat budget.

*Data availability.*

All data used in this study are published and freely available in PANGEA or the Polar Data Center. This study's model outputs will be archived and publicly available once the peer-review process is finalized.

*Author contributions.*



PI processed the data, carried out the analysis, and wrote the original manuscript draft. GEL developed the numerical modeling tools, prepared the weather reanalysis data, contributed to the analysis, and contributed to and edited the manuscript.

*Competing interests.*

PI and GEL declare that they have no conflict of interest.

*Acknowledgements.* The field observations used in this paper were collected and published as part of the international Multidisciplinary
drifting Observatory for the Study of the Arctic Climate (MOSAiC) with the tag MOSAiC20192020. We thank all persons involved in the 2019–2020 RV *Polarstern* MOSAiC expedition (AWI_PS122_00) as listed in Nixdorf et al. (2021).

We thank Malin Johansson and Wenkai Guo (UiT) for their assistance with ordering and processing the Radarsat-2 satellite data.

PI and GEL were funded by National Science Fundation (NSF) grant #1820927 (MiSNOW). PI was also supported by Research Council of Norway grant #287871 (SIDRiFT). GEL was also supported by National Aeronautics and Space Administration (NASA) Grant
#80NSSC20K1121.



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
