# Peer review of "Combining observational data and numerical models to obtain a seamless high temporal resolution seasonal cycle of snow and ice mass balance at the MOSAiC Central Observatory"

_EGUsphere, 2024_

## Author Response (AR1)

Itkin and Liston: Combining observational data and numerical models to obtain a seamless high temporal resolution seasonal cycle of snow and ice mass balance at the MOSAiC Central Observatory

**Reviewer 1**

This article presents an application of the use of snow and sea ice models (SnowModel-LG and HIGHTSI) with assimilated observations from the MOSAiC campaign to produce a continuous time series of snow and sea ice data at the location of the MOSAiC Central Observatory. As the article discusses, although MOSAiC has many high-quality observations, the campaign nevertheless occasionally experienced unavoidable data-collection interruptions. In this work, SnowModel-LG, a model used to produce snow on sea ice, is run in a 1D configuration and is used to provide input to HIGHTSI, a 1D thermodynamic sea ice model. The result is a 3-hourly time series of simulated snow and sea ice properties which helps fill in observational gaps during the MOSAiC campaign. The residual term in the SnowModel-LG budget, D, is found to correlate well with sea ice deformation.

In my view, this work is of interest to the scientific community, and I believe that the methodology of this study is sound. MOSAiC has a suite of measurements which are very well-suited to be used as assimilation for a model in a 1D configuration. SnowModel-LG is a widely-used snow-on-seaice model with detailed representations of snow processes, and the use of HIGHTSI enables the modelling of sea ice in conjunction with snow. I find it encouraging that even after interrupted observations, SnowModel-LG and HIGHTSI show high fidelity in representing snow and sea ice conditions during MOSAiC once observational corrections are applied. This study also provides some very scientifically relevant insights relating to fine-scale climate-relevant processes, and how climate-model representations of such processes could be improved. The manuscript is well-structured and generally clearly written, and the scientific conclusions follow clearly from the results. The one outstanding point for me is the availability of the data, which has not been provided with the preprint, though I recognize that the authors have stated that it will be available following publication. There are also just some minor points where I think some additional clarification would be beneficial, which I list with my comments below.

**General comments:**

- The snow density observations are linearly fitted before being used as input to SnowModel-LG. Although I understand the necessity of such a fit (given the high spatial variability relative to the seasonal evolution, as discussed in this work) and I agree with the use of it here, I would appreciate seeing some discussion of the possible biases which may be introduced from using this approach.
- Data availability: This section is currently incomplete, please include specific references for all datasets used. Also, is the model source code publicly available?

**Specific comments:**

1. Figure 5: I appreciate the authors wanting to convey all possible information and very much understand the difficulties in presenting a variety of information in a single plot, but Figure 5 is somewhat difficult to read for me. Some of the lines obstruct the points in such a way that it is difficult to see the points themselves, particularly if they're overlapped by a dotted/dashed line while also containing a white dot. I suggest possibly removing the grid

(or moving it to the background behind the points so that it doesn't obstruct them). Possibly also the fit lines could be rendered as solid lines and moved behind the data points? Since the bulk densities calculated from the individual cutter measurements are shown, perhaps the individual cutter values could be left out and shown in a supplement instead. Regardless of other possible changes suggested here, I do strongly suggest extending the plot vertically or otherwise adjusting the plot so that the legends do not overlap any data points. Otherwise, I will ultimately leave the choice of what to do here to the judgement of the authors.

- 2. Figure 7: I appreciate seeing the time series and scatter plots here, but regarding the scatter plots, it's not surprising to me that the assimilated values correlate well with the model. However, I am curious about the performance for the melt season (non-assimilated) values. Could you provide correlations for the melt period values alone? (Or would n be too small for this to be meaningful?)
- 3. Figure 7b): Snow density in mid-late July appears to fluctuate rapidly; is this due to artefacts from intermittent periods of bare ice? Would appreciate a brief comment on this. Also, just to be clear, is this the corrected density modified by the density correction parameter (line 313-315)? I am curious how large the correction was, here.
- 4. Figure 9: Since you make reference to some of the events shown in coloured shading in Fig 8 while describing this figure also, possibly consider adding the coloured shading to this figure as well. I would also appreciate more specificity in the captions and/or legend what is observed vs. what is from the model.
- 5. Figure 10: "some observations during quiescent periods are skipped in the Nloop" could you elaborate on why was done?
- 6. Line 170: How many measurements were included in this average?
- 7. Line 183: When you say 243 measurements were selected, do you mean to say that you applied some selection criteria? Or is this just all the bulk snow density measurements in the sites you're examining.
- 8. Line 309: I know you say that the plots for other ice types are similar, but I would appreciate still seeing them, perhaps in a supplement.
- 9. Line 340: Could you clarify how 1 standard deviation of snow depth is defined here? E.g. is this one standard deviation with respect to the average over the entire season?
- 10. Line 432: The reasoning as to why precipitation observations have low enough errors for this to be detected is not entirely clear to me. Are you saying that this follows from the fact that there is a strong correlation between the derivatives of *D* and total sea ice deformation?
- 11. Line 577: If around 50-60% of *D* can be explained by deformation, could you comment on what could be attributed to what remains of this residual? In particular, do you expect it to be attributable just to error or are there additional processes not being considered?

**Technical corrections:**

- Reference section: Several of the DOIs in the references appear to have formatting errors (e.g. the doi.org URL is repeated twice), which occasionally also breaks the hyperlinks in the article.
- Line 35-36: This sentence is confusing to me as it's currently phrased, did you mean to say that the drift of the expedition is shown in Fig 1?
- Line 140: survided -> survived
- Line 185-186: either should be "SMP measurement sites." or "SMP measurements."
- Line 377: SnowModel-LG misspelled

- Line 416: "affect" should be "effect"
- Line 461: phenomena -> phenomenon

**Author's Comments**

Dear referee.

Thank you very much for your review, comments and suggestions. All your comments will be addressed, references improved and simulation data and code provided/published accordingly in the revision.

**Response to the general comments:**

Both reviewers requested more discussion on the point of the snow density in-situ
measurements. In the revision we provide more discussion also on the potential biases, as
requested here. This section however remains short and we prefer not to add more
information as supplements. We made small adjustments in the final paragraph and extended
it with the following text:

If these were real fluctuations in snow density, they would cause at the same SWE about 10 \% increase in snow depth and correspondingly lower thermal conductivity further impeding ice growth on all ice types. The amount, scatter and sampling biases of data supports our choice of a linear fit as an alternative higher-degree curve could lead to over-interpretation. A dedicated study of spatio-temporal development of snow density at MOSAiC is required, but it is beyond the scope of this paper.

• A documented version of the identical code used here was published as part of the Mower et al, 2024 paper (<a href="https://gmd.copernicus.org/articles/17/4135/2024/">https://gmd.copernicus.org/articles/17/4135/2024/</a>). We will use the same Zenodo code repository as them. In addition, we provide the model simulation results and Python analysis scripts in separate Zenodo publication.

**Response to the specific comments:**

- 1. Figure 5 which represents the above mentioned in-situ snow densities was improved as recommended: there is no more grid lines, the plot is stretched vertically and model lines are behind the data.
- 2. Figure 7 has melt period correlations included into the statistical analysis. Now we added also just the correlations for the melt period. The caption was extended with: 'In the right column correlations for all data are printed in black and correlations for the melt period are written in grey. All correlations are significant.' As visible from Figure 5, there were no density measurements taken during the melt period. This is clarified in Section 5.1.2 by: 'There were no density observations taken during the melt period.'
- 3. Figure 7b and late July fluctuations: after 7 May no more observations are assimilated and there is no bias correction used here. The fluctuation is a result between bare ice situation (zero density) and density estimated for a small amount of snow (see panels a and b that show very little snow) that has fallen on a given day (typical value 230 kg/m^3) which promptly melts away under high atmospheric temperatures soon after the snowfall.

- 4. Figure 9 has now the event shading added. Caption was extended by text 'Simulations are represented by lines and observations by points.'
- 5. Figure 10 and exclusion of the Nloop measurements: Nloop was sampled so often that it had measurements in between any weather or sea ice dynamic event. Those measurements were not used. This is now better explained by: 'Nloop development can be calculated over 23 derivatives, with 8 giving meaningless rates falling between two consecutive deformation events and simply showing no change during quiescent periods. Instead, those redundant steps and their derivatives were removed from calculations here. When using similar derivatives over sampling steps in time as for Sloop, the correlation was similarly strong (R\$^2=62\%, N=15, p=0.0016).'
- 6. Line 170: This number depends on the individual transect. For example, Sloop had about one third area covered by level ice. The entire transect had at every sampling between 1000 to 1500 measurements (depending on the sampling distance by Magnaprobe). This would result in about 300 to 500 snow depth measurements on the level ice. The mean and standard deviation from these measurements were used in this paper to guide the model simulations. This also explains the reviewer's question under point 9. We added text: 'Depending on the transect 100-500 measurements were used in such calculations.'
- 7. Line 183: These were all measurements used in this paper. Some measurements did not pass the quality control due to missing values, unique sampling locations etc.
- 8. Line 309: Here are the other location calculations: Sloop (left) and Runway (right).

They are very similar to Nloop and we prefer not to include them to the paper.

- 9. Line 340: This is now explained under point 6 above.
- 10. Line 432: We are convinced about the quality of the precipitation data because of the analysis done by Matrosov et al 2022 (basic assumption in line 62) AND because the difference is small, has interchanging sign (positive and negative bias) and that coincides with how deformation processes could affect the local snow mass balance. The bias correction is very small after slightly modified approach (resulting in identical snowfall forcing) as described by text: 'For precipitation, we performed the precipitation drizzle

adjustment described in \cite{liston2020} following the trace precipitation analysis of \citep{boisvert2018}. In this procedure, the daily clipping threshold was set to 0.15 mm 3-hours\textsuperscript{-1} (1.2 mm day\textsuperscript{-1}) to create discrete, storm related, precipitation events. Then the clipped precipitation was added to the remaining nonzero precipitation periods, thus conserving the total, temporally integrated, MERRA-2 precipitation quantities. The resulting precipitation time series was then bias-corrected using the KAZR precipitation observations.'

11. Line 577: We expect that deformation is the last major missing process. Measurements of D here are spatially averaged. This is why we do the quantitative explanation about the leads shear zones from the ship radar images in addition. We add text: 'This is further supported by the analysis of the locations of active deformation zones relative to the observed SWE measurements.' We expect more can be learned from a distributed simulation – as this is now consistently claimed in the paper.

**Additional change:**

We have removed all snow depth observations after snow melt date also for the coring site snow depth observations. They do not represent snow depth, but 'surface scattering layer' depth. Now Figure 9 was modified (summer measurements from coring sites were removed) and Methods text was modified to:

Because during ice melt the ice surface is soft and granular, **any stake** measurements **(including Magnaprobe)** can erroneously detect melted ice surfaces as snow after melt onset \ citep{webster2022,itkin2023}, no **snow depth** measurements were used after mid-July.

Thank you for all the technical corrections, they were all applied!

Thank you also for your work and time. Your comments have greatly improved this manuscript. On behalf of both co-authors,

Polona Itkin

Itkin and Liston: Combining observational data and numerical models to obtain a seamless high temporal resolution seasonal cycle of snow and ice mass balance at the MOSAiC Central Observatory

**Reviewer 2**

In this paper, the authors combine the MOSAiC observational time series with two one-dimensional models, one for the snow and one for the ice, to bridge gaps in the time series and to complete one full annual cycle. The produced data set of SWE, snow density, snow and ice thickness can be interesting for other applications and from the analysis some (first) conclusions can be drawn about the impact of snow deformation on the snow depth evolution and on how snow depth heterogeneity influences heat transfer through the ice.

While the paper is clearly written in the sense that you can easily read and follow the sentences, I had often difficulties to find out what EXACTLY has been done and why. This is reflected in the numerous comments you will find below, which also might suggest that a final consistency check/internal review would have been useful and should not be task of the reviewers...

My main concerns are listed below, while individual comments follow after that:

- 1. The authors stress how important the initial conditions (for freeze up/snow accumulation start date and ice thickness) are for the simulations but the freeze up date is determined inconsistently. For the first ice type the 10m (why not 2m?) 3 hourly air temperature is used, while for the second ice type the 3-day running mean is used. However, this is not stated or discussed anywhere, I inferred it from the figure (Fig. 4). Where the assumed initial ice thicknesses, especially for the second ice type (10cm), comes from is not clear. This combined with the uncertainty from using reanalysis precipitation multiplied by 2.13 (?, see 3. point below) for the first 2.5 months of the simulations should be more appreciated as a bigger uncertainty of the produced time series.
- 2. A linear fit to the observed snow density data is assumed, which I am not convinced of. The implications and resulting uncertainties of this assumption should at least be discussed more extensively.
- 3. To drive the snow model with atmospheric data especially in the initial and final phase of the simulations, the authors use MERRA-2 reanalysis data. The precipitation from MERRA-2 is multiplied by 2.13 to do so and the authors claim that this is the result from comparing in situ precipitation observations with the reanalysis data were available, but the comparison is not presented, the high value of 2.13 is not discussed anywhere, and the precipitation in the time frames when only the reanalysis data is used is considerably higher than during the observation times (see Fig. 6c and my comments for Sect. 3.7 and Tab.1)! This raises some questions which are not addressed at all.
- 4. I suspect that something has been mixed up in Sect. 5.1 and 5.2. My guess would be that 5.2 should be given before 5.1, otherwise I would have even more difficulties to understand what has been done here (see the specific comments below). This caused lots of confusion. Because my comments would be different if or if not this is the case, I prefer this to be checked first. Anyways, I would prefer to see what the results for the simulations are with and also without the assimilations. If I understand it correctly, this is only shown for SWE in Fig. 8, and then the accordingly assimilated SWE is shown in Fig. 7a (?). I assume that in Fig. 7b and c only the assimilated snow density and snow depth are shown? It may be useful to also discuss what happens in the 'model world' when the model is dragged to match the observations (maybe strange things happen when the model is forced to use values that are not consistent within the model physics?)

- 5. Different numbers of observations are used at different locations of the paper, and I cannot always follow where these come from.
- 6. The authors claim that their analysis confirms the good quality of the precipitation data because the difference between the simulated and observed snow evolution is (relatively) small and can partly be explained by deformation. I would suggest to state more clearly that this assumption (if I understood this correctly) is mainly drawn because a correlation between the difference between simulated and observed snow evolution/SWE (which is hypothesized to be attributed to deformation) and total deformation obtained from buoy position measurements is found (r2=0.58). This chain of arguments should be stated more clearly (if this is what the authors meant).
- 7. The authors should differentiate more on what are outcomes from the MOSAiC observational timeseries (published already earlier) and the NEW insights due to combining the observations with models. I suspect that parts of the conclusions contain information related to the first and not the latter one (see specific comments below).

(Specific comments are listed just in the response.)

**Author's Comments**

Dear referee.

Thank you very much for your review, comments and suggestions. All your comments will be addressed, references will be improved, and the simulation data and code will be provided / published accordingly in the revision.

Response to the main concerns:

1. Uncertainty of the initial conditions (freeze-up date and thickness) and the atmospheric conditions: All the temperature observational data in the paper are 10-m temperatures. MicroMet model is used to simulate the 2-m temperatures that are used further in the observations. We have checked and this is consistently explained in the paper. The reason the different temperature criteria choice for the deformed and ponded SYI is the nature of the freezing point. The first has a fresh surface without any contact with the sea ice water and we assume it freezes at zero. The second is salty and connected to the sea water; we tracked that with the 3-day running mean. This not more explicitly pointed out by adding:

'the melted-through melt ponds with sea water in them \citep{macfarlane2023a}, and correspondingly low freezing point, likely began to freeze and snow started to accumulate.'

The importance of the atmospheric forcing bias is also be better explained in the revision (see point 3 here).

The choice of initial ice thickness is also better explained. We have added information about the rotten ice into the Section 'Sea ice types at MOSAiC':

'The remaining ice between ponds was 'rotten ice' honeycombed by water before freeze-up.'

And provided explanations in Section 'Initial sea ice thickness and snow accumulation':

Predominantly level and ponded SYI was 0.1 m thick on 30 August. This ice thickness was estimated based on the zero thickness in the melt ponds and ice content of the rotten ice (modal thickness approximately 0.3 m end of October).

Predominantly level FYI was initiated with 0.05 m thickness (**minimum sea ice thickness allowed by numerical model**) on 20 October.

2. Linear trend in snow density: using a linear trend is the most simple approximation. We believe this is sufficient for this study. Seasonal biases and measurement limitations are already discussed briefly. It is clear that a separate spatio-temporal study of those measurements would be required, but it beyond the scope of our paper. To explain that and potential biases arising from the liner fit simplification, we revised the text in the last paragraph of this section added at the end:

If these were real fluctuations in snow density, they would cause at the same SWE about 10 \% increase in snow depth and correspondingly lower thermal conductivity further impeding ice growth on all ice types. The amount, scatter and sampling biases of data supports our choice of a linear fit as an alternative higher-degree curve could lead to over-interpretation. A dedicated study of spatio-temporal development of snow density at MOSAiC is required, but it is beyond the scope of this paper.

3. Atmospheric precipitation bias correction: First, we need to explain how the 2.13 precipitation correction was obtained. There were two steps to the precipitation correction we implemented. 1) the "drizzle clipping" described by Liston et al. (2020) was performed, but without the mass-balance correction. Then 2) the bias correction described in the manuscript was implemented. This totaled a correction of 2.13. If the mas-balance correction is removed from the bias correction, then the precipitation bias correction is 1.01. This is now clarified in the manuscript with the text:

For precipitation, we performed the precipitation drizzle adjustment described in Liston et al. (2020) following the trace precipitation analysis of Boisvert et al. (2018). In this procedure, the daily clipping threshold was set to 0.15 mm 3-hours-1 (1.2 mm day-1) to create discrete, storm related, precipitation events. Then the clipped precipitation was added to the remaining nonzero precipitation periods, thus conserving the total, temporally integrated, MERRA-2 precipitation quantities. The resulting precipitation time series was then bias-corrected using the KAZR precipitation observations.

Then, we revised the discussion Section 'Missing time periods during the MOSAiC observation year' in respect to the bias-corrected reanalysis in early winter prior to MOSAiC in-situ observations (August and September): it is true that the difference between snowfall and SWE on the sea ice are forming about 1/3 to 1/2 of accumulated difference (see Figure 8, specially for the Nloop case), but 1) our bias correction results in realistic snow depth, snow density and ice types for all ice types, 2) even larger difference in snowfall and SWE occurs from February to May when the atmospheric forcing is exclusively from MOSAiC weather observations. We rearranged parts of discussion for clarity and added a sentence:

In particular by the onset of MOSAiC observations, the difference between accumulated snowfall and the SWE was already nearly one half.

Based on the early winter difference between snowfall and SWE on sea ice we give recommendations on earlier onset of seasonal observations in future experiments and extend a sentence in conclusions to:

Future snow and ice evolution studies, similar to MOSAiC, would benefit from actual measurements of **sea ice, ocean, and atmosphere** conditions during the freeze-up period.

4. Sequence of SWE calculations: The sequence of snow calculations is now better explained and the section 5.1 is reorganized to follow the steps in calculations. Section 5.2 is now a subsection of 5.1 (5.1.1) and snow depth calculations from SWE are described in subsection 5.1.2.

All simulations results that the reader requires are already shown on Figure 8. There is the snowfall (where no SnowModel mass balance sinks have been used), then there is 'model without D' and finally 'model with D'.

5. Problematic observation numbers: The observation number for example Nloop are not the same in all cases. Here we miscounted and there should be 15 (not 16) derivatives for Nloop. Their choice is now explained in extended text:

Nloop development can be calculated over 23 derivatives, with 8 giving meaningless rates falling between two consecutive deformation events and simply showing no change during quiescent periods. Instead, those redundant steps and their derivatives were removed from calculations here. When using similar derivatives over sampling steps in time as for Sloop, the correlation was similarly strong (R\$^2\$=62\%, N=15, p=0.0016).

In sea ice thickness correlation, all measurements are used, not just one per deformation event. There is also one more per each location as these are not derivatives. This results in N=44=17+24+3, for Sloop, Nloop and Runway.

6. Justification of good quality precipitation data: Your explanation is in line with our message, which was revised to (modified text in Conclusions):

High quality precipitation data collected at MOSAiC led to simulations with a relatively small residual, or difference between the simulated and observed **SWE**. We found that this residual term can, in large fraction (**R**^2=0.58), be explained, in a statistical sense, by the regional scale deformation that occurred over the broader area where the deformation observations were collected. **This is further supported by the analysis of the locations of active deformation zones relative to the observed SWE measurements.**

7. Differentiation between what is based on the old observational paper and what is based on the simulations here: We revised the paper and make more clear which results originate from simulations and which from observations. Especially Section 6.4 was improved in this repspect.

**Response to the specific comments (in blue):**

l. 10: 'D ... and was at times as high as 10% of all winter snowfall' -> When only reading the abstract, this sentence is hard to understand correctly. Maybe something like 'deformation appears to contribute/explain 10% of ...'.

**Changed to:**

"... and deformation appears to explain as much as 10\% of all winter snow water equivalent."

l. 28: roles of snow... has

Corrected.

l. 45: I suggest: ... collected once a week, the following caused interruptions of up to 2.5 months: list of things -> would be much easier to read. What do you mean with 'early fall MOSAiC ship arrival' and 'summer ship departure'? Sounds like the starting and ending point of the time series? Do you mean that there is a gap because the time series does not cover the full 12 months?

**Rewritten to:**

While the measurements were generally collected once a week, **late ship arrival**, **late freeze up of the melt ponds and FYI**, **crew exchanges**, **and sporadic weather and sea ice deformation events**, caused discontinuities in the observation time series of up to 2.5 months.

l. 72: 'This implies that no snow had accumulated on it during summer.' -> Do you mean: we do not know whether snow has accumulated on the ice DURING summer but at the end of summer all precipitated snow had been transformed to ice/frozen melt ponds?

This information is a speculation that is not relevant here. The sentence was removed.

l. 84: that -> which (also elsewhere)? and 'are not generally' -> 'are generally not'?

Corrected.

l. 139: '...was 0.5 m thick on 1 August': Ok, you argue that October ice thickness is a good estimate for end-of-summer ice thickness, does this mean that 0.5m ice thickness was the mean/modal/? ice thickness as measured on this ice type in October? (I had to speculate as this is not explicitly stated here)

**We added statement:**

'This ice thickness was estimated based on the modal sea ice thickness in October.'

Fig. 3, legend: 'Runwy'

Corrected.

l. 140: survided

Corrected.

l. 143: Why did you use 10 m air temperature from reanalysis? (and not 2m as it would be available for reanalysis data) I would state already here what reanalysis you used (and not only refer to section 3.7)

We prefer MicroMet to simulate 2-m air temperature. MicroMet is also used to simulate other boundary layer variables that are consistent to each other. We have checked that this consistent in the paper and that it is always stated that the air temperatures from observations at 10-m and not 2-m. The final paragraph in the section was modified to:

'All the 10-m meteorological observations described above were used as forcing in a high resolution atmosphere model MicroMet \citep{liston\_elder2006b}. MicroMet simulations also generated additional atmospheric forcing data, including cloud cover, incoming solar radiation, incoming longwave radiation, and surface pressure.'

Caption of Fig. 4:

1. the caption should include what temperature time series is shown (biased corrected MERRA2 reanalysis)

**Modified to:**

'10-m air temperature **from bias-corrected reanalysis** from 1 August through 1 September 2019'

2. sea ice water -> sea water?

Corrected to 'sea ice'.

3. 'The dates when air temperature AND its running mean depart continuously from the freezing temperature...' -> inconsistent/confusing choice of the two dates (blue and purple vertical bars): if you choose to set the dates according to the running mean, the second date agrees with this definition but the first date is not detectable from the shown time series (running mean is below the threshold for the whole tine series shown here). However, when making the choice based on the air temperature series (instead of the running mean), the first date agrees with this definition, but the second date would be shifted to a later day (and from the figure it would not be obvious whether it should be even later or not). Or if you meant to refer to this difference by using 'respectively' in the end, this is not clear and not distinguishable for the reader and would in addition require further explanation why these different definitions are used.

**The text was expanded to:**

'The date when air temperature departs continuously from the freezing temperature of water is identified by a blue vertical bar. The date when the air temperature running mean departs continuously from sea ice freezing point is identified by a purple vertical bar.'

l. 146/150: Where do these assumptions (0.1 m/0.05m thickness) come from, especially the 10cm assumption?

Both assumptions are now explained. See also major concern 1.

l. 147: bellow

Corrected.

Section 3.2 + 3.3: Maybe it would be better to combine these sections as these paragraphs are a bit confusing because they jump back and forth from the different measuring methods (transects, drilling sites, stake sites) without clear transitions.

The sections were merged and arranged by method/locations as suggested.

l. 156: 'no Magnaprobe snow depth measurements were used after mid-July' -> more precisely (I guess): after mid-July 2020 (i.e. the final phase of MOSAiC)

**Year added.**

Section 3.4, l. 174-192.: the jumping back and forth between the different methods to determine snow density makes this section harder to read 1. you mention all three methods (ok), then you write something about the cutters, then about the SMP, then about the SWE cylinders, then again the cutters, the SMP, the SWE (I suggest to go through this cycle once, this also avoids unnecessary repetitions).

The first 3 paragraphs in the sections have been rearranged so that each of them is dedicated to one of the 3 methods.

l. 183: why 'still'?

Removed.

l. 219: 'In addition, the final sampling on Snow 1 provided deeper and denser snow values.' -> meaning what?

**Modified to:**

'The final sampling on Snow 1 provided again deeper and denser snow values coinciding with the linear fit.'

Fig. 5:

1. you give N=591 but maybe you could add in the caption the numbers for the three different methods such that the reader does not have to search them from the text.

Sample numbers were added to the caption.

2. fit line for cylinder measurements: in spring 2020 there seem to be enough measurements (marked by crosses) but all of them are far below the fit line. It is hard to judge from this figure (with all the others points for the other two measurement methods) whether the fit is mathematically correct (which it probably is) but it also shows very clearly that a linear fit as calculated here is not a very feasible assumption for these data points... (which should at least be mentioned somewhere)

We discuss the spatial representation problem of these measurements (due to restricted access during break up). In the revision this part has an extended discussion as addressed under major concern 2.

3. equation rho\_s = 0.22 x is given without units (also in the text description)

The units were added ( $kg/m^3$ ).

Section 3.5: for the other sections you already show some numbers/results, here not. Are the shaded areas, e.g. in Fig. 6, determined here?

Subsection 3.5 got an additional explanation of its purpose by the statement:

'To perform our year-long snow and ice simulation, we required ice parcel location information during the entire simulation year (1 August 2019 through 31 July 2020). **Only then could the atmospheric reanalysis data be extracted.**'

This subsection is referring to Figure 1. The result is minor, stated in the last sentence and now extended to:

'As shown on Figure \ref{fig1} all used drift trajectories coincide well with each other **and the reanalysis data are the same for all trajectories**.'

We have reordered the sequence of chapters in Section '3 Observations used', so that we start with the location and then run through the vertical column of observations from atmosphere to ocean. This helps with the explanation of the shading on the figures that comes towards the end of the subsection 3.5:

'Based on the weather and sea ice deformation data, we determined 4 periods, when high winds, snowfall, high deformation rates and large relative motion in the MOSAiC CO (Figure \ref{fig3}) were observed. These periods and marked on Figures \ref{fig6} and \ref{fig8} and will be used to discuss the deformation-associated snow sinks. The periods are: \begin{enumerate}

\item October shearing and ridging from 16 October to 1 November: marked by blue shading. \item November shearing and ridging from 14 November to 5 January: marked by yellow shading.

\item January lead from 22 January to 7 February: marked by pink shading. \item March-April leads from 15 March to 30 April: marked by green shading.'

We shortened the captions of Figures 6 and 8.

In subsection 6.2 (Discussion) we expanded text:

To understand the reasons for these statistical correlations, we tracked the timing and location of major leads and pressure ridges (Figure \ref{fig3}) occurring in the vicinity of the MOSAiC transects on the ship radar images (see Section \ref{sec\_buoys}) **during periods of increased winds and snowfall, and high deformation rates (Figures \ref{fig6} and \ref{fig8})**.

Section 3.6 + Fig. 1: Is the 'buoy' given in Fig. 1 the buoy 2019I3 mentioned in the text? For me, from the description it is not clear why there are different drift trajectories and what the different trajectories are: So, in the figure, the trajectories marked as MOSAiC

CO1/CO2 are from the ship positions? and they are almost identical to the buoy, right? Reading the text, I would assume that the back trajectory model is only used to extend the ship's or the buoy's trajectory at the beginning and the end, but the figure suggests that this is a separately derived drift trajectory for the whole time series. Wouldn't it be better to use the buoy's trajectory where available and only to add the beginning and the end? Also, the beginning of the trajectory looks strange...

This figure shows how very similar is the trajectory of the coarse passive microwave satellite-based (PMW) sea ice drift that was used to extract the atmospheric reanalysis data from August 2019 to August 2020. The distance between trajectories is so small that this does not influence the selection of the MERRA-2 forcing (both tracks fall withing the same model grid cell for any point in time). The beginning has no buoy as none was deployed and only 'strange PWM drift is shown'.

This was now modified as documented in the previous point.

Section 3.7 and Tab. 1: It would be interesting to see the MERRA-2 data compared to the observations. The authors multiplied the water equivalent precipitation from MERRA-2 with 2.13 and (as a result?) in the time sections where the reanalysis data is used, the precipitation is considerably higher than in the remaining time: increasing from 0 to 5m within less than 2.5 months and later by approx. 3m in 2.5 months (of reanalysis data), while the cumulative precipitation increased by only approx. 4m in the remaining more than 7 months (of observation data). The reader should a least have a possibility to retrace whether this is realistic and where this comes from and what uncertainties are involved.

This was now clarified as described under under major concern 3.

l. 260: I suggest to write: 'In this paper, only ocean surface heat fluxes derived from SIMBA buoys

by Lei et al. (2022) were used.

The sentence was removed and statement 'Several other researchers have used SIMBA or similar data to estimate snow depth and sea ice thickness evolution at MOSAiC \ citep{lei2022,perovich2023,salganik2023b}.' was moved to the end of the section.

l. 275: SWE is calculated -> the change in SWE (with time) is calculated

The text was modified to 'the change in SWE with time is calculated'

1. 280: the units ... is -> the unit ... is / units ... are

Corrected.

l. 304: 'any and all'?

Simplified to 'any'.

Section 5.1: The description in this section could be clearer. I am also not convinced that a linear fit to the snow density values (Fig. 5) is the best choice to describe the observed snow densities and whether this fit is something you would want to use to correct/assimilate the model. Again, I would like to see how the model and the observations (e.g. of snow density) differ when the model is run without assimilations? And if there is large differences the reasons for this should be discussed somewhere. And what happens in the model when snow density and thickness are just changed (assimilated) without physical reasons (in the model's world).

The description of this section and the snow density data was revised and improved as described under major concerns 2, 3 and 4.

Fig. 7: Do I understand correctly that in the scatter plots the assimilated model values are compared with the observations (and thus are located (almost exactly) on the 1:1 line and only because the model is not assimilated for the melt season, there is some scatter points where the model and the observations differ (which then lead to r2 values not equal to 1.0). I am not sure how much sense these scatter plots make under these circumstances...?

Yes, you are correct. This figure clearly shows two things: that our assimilation methodology works and was done appropriately, and that the model physics realistically describes what happened during the melt period when the observations were not assimilated. This understanding is critical for the reader to accept the general findings of the paper.

I guess the circles are not only 'assimilated values' (purple) and 'melt period values' (grey) but could be marked as 'observed values' (which are used for assimilating the model (purple) or not (grey, melt season)).

Yes, these circles are observed values. The caption text was modified to:

- 'Assimilated observed values are represented by purple circles. Not assimilated observed values (melt period) are represented by grey circles.'
- l. 310: 'the final SWE evolution defined above was further modified' -> what is the 'final' evolution and were 'above' is it defined and what do you mean with 'further' modified, what is the first modification?

This now more clear after the merge of the section and revision described under major concern 4.

l. 320: Snow depths were created using eq. 2 (or 3?) and the assimilated SWE and snow density values, but they were also (on top) assimilated with observed snow depth values?

Both equations (finally 3, of course).

Now all equations in the paper are labeled and referenced in the text where appropriate. This also includes equations for strain rates, snow density linear fit and snow thermal conductivity.

The assimilation procedure is now revised according to modifications under major concern point 4.

Section 5.2.: The modifications to SWE described here are they meant in l. 310?

This is revised as explained under the major concern 4.

Fig. 8: Caption does not explain all markers used in the figure. It is confusing that the SWE observations are (blue and gray) crosses (I am guessing, it is not explicitly stated in the caption) while the other crosses are the derivatives. Why are the absolute precipitation values so different for the three ice types? Are they only shown qualitatively, do they refer to the values on the y-axis?

The markers are explained on the legend. The accumulated precipitation differs by the start of the accumulation date.

l. 339: strange sentence: snow depth variability ... snow-depth variability?

**Corrected.**

l. 340 f.: Did you come up with the idea to use mean snow depth minus 1 std or has it been used elsewhere earlier? Is this related to the fact that the effective thermal conductivity of snow (partly due to snow depth being variable instead of one mean value) would be much higher than the thermal conductivity (point) measurements that you are using here? (edit: you make this connection much later in the paper)

However, if only 16% of the data show this or a lower value, it sounds like a quite small value to be used for the simulations? Maybe this is more like 'tuning' the ice thickness model (for Nloop and Sloop) and there are other reasons why the simulated ice is otherwise thinner than the observed one?

We added a statement into the Discussion (Section 6.3):

'Further spatially distributed, 3-dimensional, simulations of snow and sea ice are necessary to further study the heat fluxes and local consequences of snow for sea ice growth.'

1. 378: than than

**Corrected.**

For me, what is written in Section 6.1 is a summary with some conclusions but not a discussion...

We agree with the reviewer to some degree, but the logical organization of the paper as it will enable the reader to select one of the 4 results and read the information on it (summary and or

discussion). Complete time series simulations will published with this paper and are hopefully going to be used in sea ice and interdisciplinary studies.

This section was modified according to the major concern 3.

l. 394: 'SnowModel-LG reproduced the observed snow evolution' because it was assimilated

Yes, this was a key goal of the work: to reproduce the observations when and where they occur, and use the physics in the modeling system to provide appropriate data values between those observations.

l. 400: 'These SSS and SBS values were about three times as large as in Liston et al. (2020)' -> or maybe this is related to the precipitation in the reanalysis being multiplied by 2.13?

The reasons for this are discussed right after the statement. Again, we would like to point out that the atmospheric forcing was for most of the period from the MOSAiC weather observations and not form the reanalysis. Reviewer is right at the point that about 1/3 of the difference between precipitation and model without D originates from the period in September (with large snowfall), but ever larger difference appears during spring snowfalls (from February on) when no reanalysis forcing is used.

According to the revised procedure, the bias correction factor is small. See also response to major concern 3.

l.414/5: This could be mentioned already on l. 327 to explain why you set D to zero for these time periods.

We clarified this assumption on lines 326 and 327 where we first discuss setting D to zero: 'Before and after these dates D was set to zero, under the assumption that no blowing snow occurs when air temperatures are above freezing and the snow may be melting (e.g., Li and Pomeroy 1997).'

1.416 affect -> effect + I suggest to phrase this more precisely: deformation contributed to ... % of ...

Corrected. We prefer to keep the text concise here.

l.426: between ... to ... -> between ... and ... + Why is there only n=16 data points for Nloop and Sloop and only n=2 for Runway? (in Fig. 7 there are more...)

This was rewritten as explained under major concern 5.

l.432/3: 'snow redistribution ... was able to be detected' -> 'we were able to ...' or 'it was possible to...' or '...could be detected'?

Changed to 'could be detected'.

l.432/3: 'Our findings confirm that the precipitation observations have sufficiently low errors that a minor signal such as snow redistribution due to sea ice deformation was able to be detected' -> With 'our findings confirm' do you mean the high correlation between the time derivative of D and the total deformation? If so, please write it like that. If not, please elaborate more on why you think the findings (which?) confirm the accuracy of the precipitation data. Because otherwise I would think that the higher the uncertainty/error in the precipitation data, the higher the fraction of snow that

would have to be (erroneously) attributed to deformation in order to make the simulations and the observations match.

This was addressed under major concern 6. Here we additionally shortened the text to: 'This analysis was only possible...'

l. 461: 'a known phenomena' -> phenomenon

**Corrected.**

l. 480: check sentence: for example, what is 'younger ice thickness' + 'the oldest and thickness ice'

**Corrected to 'FYI'.**

Fig. 11: Why are there only 3 data points used for Runway? Why are the ice thicknesses from the coring site and the Ridge Ranch representative enough to be included in the comparison in Fig.9 but not here? If they were included, the correlation would be smaller because these observations show clearly thinner ice than the simulations with snow depth = level ice mean minus 1 std.

This is discussed elsewhere in the paper: There are 3 different ages of FYI, each resulting in different SWE and snow depth. Here the simulation is only done for the Runway (that has enough measurements to provide standard deviation). Other measurements (from stakes and coring) are provided to give a supporting information, but can not be used for quantitative analysis. Again, one of the weaknesses of the MOSAiC dataset is poor sampling of the FYI (due to logistic restrictions).

This information is already included into the paper and discussed just before the statistical analysis. To make this more clear we added at the end of the section:

'Another weakness of the MOSAIC dataset is lack of repeated transects on the FYI (as discussed in Sections \ref{transect} and \ref{point}). The data from the coring sites and stakes are not directly comparable to the transect location (see also discussion above) and can not be included into the statistical analysis (Figure \ref{fig11}).'

l. 499: Why not write 'Our bedform parametrization (using the level ice mean snow depth minus one standard deviation),...' instead of 'Our bedform parametrization, as described in Section 5.3,...'? Easier for the reader, only a few letters more.

**Description was added as suggested.**

l. 499-505: Interesting idea, but I think a more detailed and more comprehensive analysis focusing on this aspect should be conducted before any recommendations can be given of what works better (using a different thermal conductivity value vs. a different snow thickness value (and especially which one))

This is a comment about the 3-d heat fluxed through the ice. On this point we disagree with the reviewer as this is the fourth publication the idea is raised. If reviewer insists, we can remove this part.

l. 500: that -> than

Corrected.

1. 507: citation not correctly embedded

Katlein et al, 2020 is correctly embedded in the PDF we submitted, we will pay attention to this and other citations.

l. 509: I assume you want to refer to Section 3.1 also for the initial thicknesses.

Correct, reference to section 2.1 will be removed.

Section 6.4: I would suggest to use consequently 'the simulated snow/ snow density/SWE/whatever' to make it clearer when you write about the simulated results in contrast to observations

This was rewritten as suggested.

l. 531 f: 'All [simulated, see comment above] snow melted by 8 July, which fits well with the transect observations ... On level ice with snow reduced by one standard deviation, [simulated] snow was fully melted even earlier - about 3 weeks prior to the average snow cover. This coincides well with the estimates from eight thermistor chains deployed on level ice (Lei et al., 2022).' -> What does this mean? According to transect observations (i.e. real people on the ice) the ice melted on 8 July, according to (eight) thermistor chains this happened already three weeks later? Are eight thermistor chains only a subset of all (how many?) thermistor chains? Is this related to spatial heterogeneity/a different ice type?

This part was simplified to:

'All simulated snow melted by 8 July, which fits well with the transect observations \citep{webster2022,itkin2023} and thermistor chains \citep{lei2022}.'

l. 544/5: What do you mean? There is more melting around thermistor chains?

Yes.

l. 556: Again, I guess you mean Sect. 3.1?

Corrected.

l. 559-567: Are these results obtained from combining the model and the observations or are/can they (be) obtained from the observations? (and have been published before) The newly added information through your approach of combining the models with the observational time series should be made visible here.

The list of conclusions here is based on the simulations in this paper. Some findings coincides with the findings from other MOSAiC publications (but this is always supported by a citation as usual). What is important here that the simulations unite all these findings that are otherwise disconnected and scattered in several papers. This is not clarified by stating:

**'Our simulations results from** the data-model fusion methodology'

l. 579: is the parametrization really found in 5.1?

As of now, this is in section 5.1.1

l. 594: allow -> allow for?

**Corrected.**

**Additional change:**

We have removed all snow depth observations after snow melt date also for the coring site snow depth observations. They do not represent snow depth, but 'surface scattering layer' depth. Now Figure 9 was modified (summer measurements from coring sites were removed) and Methods text was modified to:

Because during ice melt the ice surface is soft and granular, **any stake** measurements **(including Magnaprobe)** can erroneously detect melted ice surfaces as snow after melt onset \ citep{webster2022,itkin2023}, no **snow depth** measurements were used after mid-July.

Thank you very much for your work and time. Your comments have greatly improved this manuscript.

On behalf of both co-authors,

Polona Itkin

---

## Referee Report (RR1)

First of all, I would like to express that I appreciate that the authors have worked on their manuscript! I think the structure and thus readability have really improved, and it is easier to understand what has been done here. The motivation behind and main idea of the paper are of course still very relevant and would serve the scientific community. And I think that the continuous year-long snow and ice time series produced here would be really useful. However, I think that the authors should either focus on producing the most realistic estimation of a continuous snow and ice time series assimilating the model to the observations and "do the best they can" to achieve this OR focus on "praising" their model and show how well the simulations agree with observations (see my comments 1 and 3) . Here, it looks a bit like they are trying to do both... In addition, there are still inconsistencies (see comment 2), and it is still unclear to me which of the described observations have been used (and which not) and why (comment 4). As I think that most of these concerns can mainly be addressed by modifying the description instead of the actual work, I would consider these minor revisions.

- 1. The authors (still) claim that the SnowModel LG reproduces the snow evolution accurately (p. 21, l. 309-401 and l. 420-421\*¹ and p. 28, l. 588-589) during the time periods where there are no observations. The difference D (in terms of SWE) between the simulations and the observations has been subtracted from the simulations to match the observations (especially important for the two SYI cases), so of course the time series agrees well with the observations but we do not have information for the time periods in between, so how can the authors judge on this...? An exception is the good agreement between simulations and observations in June/July, which were not assimilated, which is why here the model can be "evaluated" (as is separately mentioned on p.21, 402-405). But note also my comment 3 as the starting point of the model might have been adjusted to potentially match these observations (or the ice thickness observations) better (?).

  \*¹Additional remark: I had already mentioned this in the first review round, pointing at the sentence that is now found on l. 420-21 (before: l. 394) and the authors replied "Will be corrected", which was obviously not done as the exact same sentence is still in the manuscript.
- 2. The authors write that D (= the difference between observations and simulations, assumed to be related to deformation) is small throughout the simulations (p.22, l. 443) and that "its accumulated effect by 7 May (the last winter observation) was 10% in Nloop". I guess they mean 10% of the precipitated SWE value (which still means that the simulated SWE at this point in time has to be reduced by  $\sim$ 50% to match the observations). However, based on Fig. 8 it should be  $\sim$ 15% (visual inspection)... The number of 10% is also given in the abstract, should be checked.
- 3. The authors did not explain (although I asked for it in the first review) why the surface freeze-up date (= start of snow accumulation) for the fresh ice ("deformed SYI") was determined using the (3 hourly) air temperature time series, while for the saline ice ("ponded SYI") it was determined using the 3 day running mean of that time series. (In their reply they explained why different thresholds were used, which was clear (different freezing temperatures of fresh and saline ice) but not why the 3 day running mean was used in one case but not the other). If this cannot be explained, I (have to) guess it was because the results (e.g. regarding the match of simulations and not assimilated observations in June/July and/or the agreement between simulated and observed ice thicknesses) fit better when doing it like this...? I think it is ok to use a more "subjectively" determined starting date for the simulations (as compared to using identically and thus more objectively derived dates for both) but this has to be communicated. Especially, as it is the authors themselves who stress that "The correct initialization of our simulations proved to be the most critical aspect of our work." (p. 28, 1. 603).
- 4. In Section 3.4, the MagnaProbe snow depth measurements along the transects during MOSAiC are described but I cannot see where they are used in this study. In Section 5.1, the authors write that they use SWE observations (described in the section on snow density 3.5) to assimilate the

Snow Model LG model. And in a second step they use the snow density evolution over time (eq. 1, snow density is a function of days since 25 Oct 2019) to assimilate snow density. The simulated snow depths are then compared to the snow depths\*2 that result from the SWE observations and this snow density evolution (snow\_depth = SWE \* density\_water/density\_snow) and NOT to any (directly) observed snow depths. Thus, I guess snow depth observations have not been used for SnowModel LG simulations or assimilations (?). Later on, the authors write that "Our simulations used observed MOSAiC snow depths ... to drive our ice growth simulations" (p. 19, l. 370-371). Here, it sounds like (directly) observed snow depths have been used, but this would contradict the statement a few lines earlier, where it says that "we performed ... SnowModel-LG simulations ... to create the snow forcing for the HIGHTSI sea ice simulations... using SWE observations" (p. 19, l. 359-362). Only in Fig. 9c (directly) observed snow depths are shown, but only the ones from coring and the "Ridge Ranch". Are the snow depths measured along the transects with MagnaProbes shown or used anywhere?

Another aspect that is unclear to me: The SWE observations are introduced in section 3.5 ('Snow density') as being measured in the snow pits. Figures 7, 8 and 9 show time series for the three 'transects' (Nloop, Sloop, Runway). On page 6 the locations of the snow pits are given (l. 128 ff). Nloop and Runway are among these. Did you use only SWE measurements located at these transects or all of them? For Sloop, did you take the closest ones (temporally and/or spatially) or all the measurements at any time on level and ponded SYI? Wouldn't it be interesting to see how the snow depths measured along the transects compare with the simulations and the SWE-based snow depth estimations? Or maybe the snow depth observations along the transects ARE used somewhere but I cannot find it...

\*2 It is confusing that the snow depths used in this context are called 'snow depth observations', as they are rather 'snow depth estimates based on SWE observations', especially because snow depth observations have actually been collected, too.

- p. 8, l. 161: analysis of (Boisvert et al., 2018) --> remove parentheses
- p. 14, l. 272: So far, mainly "Central Observatory" was used, here you use "CO" + "These periods and marked...": and -> are?
- p. 15, eq. (3): In the equation "M" is used, while in the text "S\_M" is used.
- p.25, l. 511-512: "... led to younger and thinner sea ice thicknesses ... equal to that of the oldest and thickness ice types" -> still does not make sense to me, did you mean "oldest and thickest ice type"?

---

## Author Response (AR2)

**Reviewer 1**

Overall, the authors have addressed the review comments to my satisfaction. The changes made by the authors clarify the text and better justify the results compared to the previous revision. I just have a few specific remaining questions and technical corrections detailed below. So long as these points are addressed, I would be happy to recommend the article for publication.

Regarding the expanded discussion of the snow density measurements, I can agree with the authors that detailed examination of snow density is beyond the scope of the paper. I very much agree that further analysis of snow density is required, but this is difficult given the high variability in snow density as is mentioned. As such, I think the amount of discussion in the revised article is sufficient.

I appreciate that the authors have now included simulation data and code, since that helps with data availability. Regarding the analysis for the other locations (Sloop and Runway), having now seen the figures I can agree they are similar, and since the output data is now provided in case others are interested in these other results, I can grant that it suffices to simply show the location with the most data (Nloop) as is done in the revised manuscript. The section restructuring for the article (e.g. Section 5) also helps the article flow better.

**Remaining questions:**

- 1. I appreciate that the authors now including the correlation breakdown by melt period in Fig 7 per my previous suggestion. However, I am confused as to why the breakdown is not shown for snow density (Fig 7 b), since there do appear to be points available during the melt period (the grey dots around 2020-07 in Fig 7 b). Is this omitted because these densities are indirectly calculated from SWE? I am somewhat confused as to what the grey dots in Fig 7b represent here and would appreciate some further clarification; perhaps specify more clearly in the text where the grey dot values are obtained from.
- 2. Although the authors acknowledge my suggestions in their review response, Figure 6 (previously figure 5; the MOSAiC bulk snow density time series plot) appears to not have been updated accordingly. I assume this was just an accidental omission but would appreciate for the figure to be updated for the final manuscript as the authors describe.
- 3. The rearranged sections are clearer, but now I think it would be helpful if the authors would briefly specify in Section 3.2 which specific precipitation type provides the forcing for MicroMet (i.e. is it total precipitation or snowfall rate and rain rate from MERRA-2). Likewise please briefly specify here what precipitation type is used from the KAZR for this study.

**Technical corrections:**

Backward opening single quotes e.g. at L72 'rotten ice' (same issue at L101, 112, 280, 425, 435)

L 272: "These periods and marked" -> "These periods are marked"

L 540: Katlein et al. citation should be parenthetical, not in-line

L 590: "Simulations results" -> "simulation results"

**Author's Response**

Thank you very much for your time and expertise. Our manuscript was improved greatly by your efforts. See below the additional modifications we made in this revision.

**Question 1:**

This is a very good point. There were no observations of snow density after the snow melt onset. The value used here was at constant 550 kg/m3 that was estimated. Since this is not a real measured value nor it is used in the simulations, we have now modified Figure 7 and omitted these data points.

**Question 2:**

This is true – the figure was updated, but not uploaded into the revision. This is now updated as promised. Apologies!

**Question 3:**

KAZR data for MOSAiC (published by Matrosov et al, 2021) has only type – total precipitation. From MERRA-2 we also used total precipitation, and we let MicroMet distinguish between snowfall and rain using the air temperature formulation of Dai (2008) as described in Liston et al. (2020). This information is now briefly clarified by:

All of the MetCity, KAZR, and MERRA-2 data were then aggregated to 3-hourly values used in the model assimilations (averages for air temperature, relative humidity, and wind speed and direction, and sums for total precipitation). MicroMet was used to distinguish between snowfall and rainfall using the air temperature formulation of Dai (2008) as described in Liston et al. (2020).

Dai, A. (2008). Temperature and pressure dependence of the rain-snow phase transition over land and ocean. Geophysical Research Letters, 35, L12802. https://doi.org/10.1029/2008GL033295

We thank the reviewer for the technical corrections – they were implemented! In addition, we did some minor technical (language) corrections in the text and figures. All of them are tracked in the 'difference' PDF file.

**Reviewer 2**

First of all, I would like to express that I appreciate that the authors have worked on their manuscript! I think the structure and thus readability have really improved, and it is easier to understand what has been done here. The motivation behind and main idea of the paper are of course still very relevant and would serve the scientific community. And I think that the continuous year-long snow and ice time series produced here would be really useful. However, I think that the authors should either focus on producing the most realistic estimation of a continuous snow and ice time series assimilating the model to the observations and "do the best they can" to achieve this OR focus on "praising" their model and show how well the simulations agree with observations (see my comments 1 and 3). Here, it looks a bit like they are trying to do both... In addition, there are still inconsistencies (see comment 2), and it is still unclear to me which of the described observations have been used (and which not) and why (comment 4). As I think that most of these concerns can mainly be addressed by modifying the description instead of the actual work, I would consider these minor revisions.

1. The authors (still) claim that the SnowModel LG reproduces the snow evolution accurately (p. 21, l. 309-401 and l. 420-421\*1 and p. 28, l. 588-589) during the time periods where there are no observations. The difference D (in terms of SWE) between the simulations and the observations has been subtracted from the simulations to match the observations (especially important for the two SYI cases), so of course the time series agrees well with the observations but we do not have information for the time periods in between, so how can the authors judge on this...? An exception is the good agreement between simulations and observations in June/July, which were not assimilated, which is why - here - the model can be "evaluated" (as is separately mentioned on p.21, 402-405). But note also my comment 3 as the starting point of the model might have been adjusted to potentially match these observations (or the ice thickness observations) better (?).
\*1Additional remark: I had already mentioned this in the first review round, pointing at the sentence that is now found on l. 420-21 (before: l. 394) and the authors replied "Will be corrected", which was obviously not done as the exact same sentence is still in the manuscript.

- 2. The authors write that D (= the difference between observations and simulations, assumed to be related to deformation) is small throughout the simulations (p.22, l. 443) and that "its accumulated effect by 7 May (the last winter observation) was 10% in Nloop". I guess they mean 10% of the precipitated SWE value (which still means that the simulated SWE at this point in time has to be reduced by  $\sim$ 50% to match the observations). However, based on Fig. 8 it should be  $\sim$ 15% (visual inspection)... The number of 10% is also given in the abstract, should be checked.
- 3. The authors did not explain (although I asked for it in the first review) why the surface freeze-up date (= start of snow accumulation) for the fresh ice ("deformed SYI") was determined using the (3 hourly) air temperature time series, while for the saline ice ("ponded SYI") it was determined using the 3 day running mean of that time series. (In their reply they explained why different thresholds were used, which was clear (different freezing temperatures of fresh and saline ice) but not why the 3 day running mean was used in one case but not the other). If this cannot be explained, I (have to) guess it was because the results (e.g. regarding the match of simulations and not assimilated observations in June/July and/or the agreement between simulated and observed ice thicknesses) fit better when doing it like this...? I think it is ok to use a more "subjectively" determined starting date for the simulations (as compared to using identically and thus more objectively derived dates for both) but this has to be communicated. Especially, as it is the authors themselves who stress that "The correct initialization of our simulations proved to be the most critical aspect of our work." (p. 28, 1. 603).
- 4. In Section 3.4, the MagnaProbe snow depth measurements along the transects during MOSAiC are described but I cannot see where they are used in this study. In Section 5.1, the authors write that they use SWE observations (described in the section on snow density 3.5) to assimilate the Snow Model LG model. And in a second step they use the snow density evolution over time (eq. 1, snow density is a function of days since 25 Oct 2019) to assimilate snow density. The simulated snow depths are then compared to the snow depths\*2 that result from the SWE observations and this snow density evolution (snow\_depth = SWE \* density\_water/density\_snow) and NOT to any (directly) observed snow depths. Thus, I guess snow depth observations have not been used for SnowModel LG simulations or assimilations (?). Later on, the authors write that "Our simulations used observed MOSAiC snow depths ... to drive our ice growth simulations" (p. 19, l. 370-371). Here, it sounds like (directly) observed snow depths have been used, but this would contradict the statement a few lines earlier, where it says that "we performed ... SnowModel-LG simulations ... to create the snow forcing for the HIGHTSI sea ice simulations... using SWE observations" (p. 19, l. 359-362). Only in Fig. 9c (directly) observed snow depths are shown, but only the ones from coring and the "Ridge Ranch". Are the snow depths measured along the transects with MagnaProbes shown or used anywhere?

Another aspect that is unclear to me: The SWE observations are introduced in section 3.5 ('Snow density') as being measured in the snow pits. Figures 7, 8 and 9 show time series for the three 'transects' (Nloop, Sloop, Runway). On page 6 the locations of the snow pits are given (l. 128 ff). Nloop and Runway are among these. Did you use only SWE measurements located at these transects or all of them? For Sloop, did you take the closest ones (temporally and/or spatially) or all the measurements at any time on level and ponded SYI? Wouldn't it be interesting to see how the snow depths measured along the transects compare with the simulations and the SWE-based snow depth estimations? Or maybe the snow depth observations along the transects ARE used somewhere but I cannot find it…

\*2 It is confusing that the snow depths used in this context are called 'snow depth observations', as they are rather 'snow depth estimates based on SWE observations', especially because snow depth observations have actually been collected, too.

Other minor comments:

- p. 8, l. 161: analysis of (Boisvert et al., 2018) --> remove parentheses
- p. 14, l. 272: So far, mainly "Central Observatory" was used, here you use "CO" + "These periods and marked...": and -> are?
- p. 15, eq. (3): In the equation "M" is used, while in the text "S\_M" is used.
- p.25, l. 511-512: "... led to younger and thinner sea ice thicknesses ... equal to that of the oldest and thickness ice types" -> still does not make sense to me, did you mean "oldest and thickest ice type"?

**Author's Response**

Here we would first like to state our gratitude for the work of this reviewer. Her/his comments were again very detailed and valuable for improvement of this manuscript. The first revision required a lot of text rewriting and some details have slipped our attention there. It is remarkable how this reviewer found many (hopefully all!) of them and pointed them out to us. It was our pleasure to be able to prepare the second revision.

**Issue 1:**

We agree with the reviewer of why the model is giving good results – in addition to the sound physics in the model, the initial conditions were carefully assessed and afterwards observations were assimilated. We carefully examined the wording in the listed statements and made small adjustments that keep the statements sufficiently short to be understandable. Here are the disputed statements one-by-one:

**OLD:**

However, the bias-corrected atmospheric reanalyses is of sufficient quality that SnowModel-LG **(apparently) accurately reproduced** these missing periods.

**NEW:**

However, the bias-corrected atmospheric reanalyses **data are** of sufficient quality that SnowModel-LG **simulated physically-credible values during** these missing periods.

**OLD:**

Using observed and estimated atmospheric forcing data, and periodic SWE and snow density observations, SnowModel-LG **reproduced the observed** snow evolution on the three sea ice types with different age and sea ice deformation characteristics found at MOSAiC (Figure 8).

**NEW:**

Using observed and estimated atmospheric forcing data, and periodic SWE and snow density observations, SnowModel-LG **simulated physically-credible** snow evolution on the three sea ice types with different age and sea ice deformation characteristics found at MOSAiC (Figure 8).

**OLD:**

Here, we have combined physics-based modeling tools, with temporally incomplete measurements, to create a full annual time series of 3-hourly snow and ice property values that match the observations when and where they occurred. Finally, the time series data contain **realistic** values when observations were not available.

**NEW:**

Here, we have combined physics-based modeling tools, with temporally incomplete measurements, to create a full annual time series of 3-hourly snow and ice property values that match the observations when and where they occurred. Finally, the time series data contain **physically-credible** values when observations were not available.

**Issue 2:**

We are very grateful for the meticulous work of the reviewer on this issue! We have recalculated the fractions of SWE removed by different sinks in comparison to the total precipitation. The recalculated results are listed below and the numbers in the text are adjusted to correspond to them. The changes are noticeable, but still small and do not alter the interpretation of the findings in this paper.

**Nloop**

May 7 cummulative effects:

Cumulative snowfall: 0.5396220300000004

model mean no D: 0.17634 model mean with D: 0.09262

fraction of SWE removed by S 0.6732157135986463 fraction of SWE removed by D 0.1551456303590866

**Sloop**

May 7 cummulative effects:

Cumulative snowfall: 0.42294580000000026

model mean no D: 0.13291 model mean with D: 0.09871

fraction of SWE removed by S 0.685751696789518 fraction of SWE removed by D 0.08086142479721982

**Runway**

May 7 cummulative effects:

Cumulative snowfall: 0.31302074

model mean no D: 0.09034 model mean with D: 0.09095

fraction of SWE removed by S 0.7113929255933649 fraction of SWE removed by D -0.0019487526609259162

In addition we get an impression that the ability of the model to simulate the SWE sinks was not explained sufficiently. The precipitated SWE is not removed to match the observed SWE. 68-70% of the SWE is simulated to be removed by a  $S_{SS}$  and  $S_{BS}$  model sink terms. These sinks are operating at the synoptic length scales (approx 100km). Only afterwards the remaining 10-1% of the excess comparing to the observed SWE is removed by our parametrization that works at LOCAL scale. To clarify also this the text has been modified as presented below. Also the value in the abstract was modified to 15%:

**OLD:**

On all three ice types, the strongest winter season sinks were static and blowing snow sublimation (\$S\_{SS}\$ and\$S\_{BS}\$) which by 7 May cumulatively removed 68, **68**, and **70** \% of \$SWE\$ from snowfall (\$P\_S\$) in Nloop, Sloop, and Runway, respectively. This is represented by the difference between 'precipitation' and 'model no D' on Figure \ref{fig8}. Note that, in this environment, if the blowing snow is not captured by a ice-topographic drift trap, or blown into an open lead, it blows perpetually and, in air that has a humidity deficit, it eventually sublimates completely away \citep{tabler1975,liston\_sturm2004}. These \$S\_{SS}\$ and\$S\_{BS}\$ values were about three times as large as in \cite{liston2020}. This is likely due to the specific weather during MOSAiC winter and location during the drift, including, generally low snowfall (\$P\_S\$) after freeze-up, frequent storms with high winds \citep{rinke2021}, and relatively high sea ice concentration \citep{krumpen2021} with low near-surface relative humidity during winter. \$P\_S\$,

\$S\_{SS}\$, and \$S\_{BS}\$ operate at synoptic temporal and length scales, and were the same (or very similar for \$S\_{SS}\$ and \$S\_{BS}\$, which depend on grain bounding) for all ice types.

The differences in \$SWE\$ evolution on the three ice types were largely controlled by the ice (and snow) onset date, and the differences in the remaining wintertime snow sink or source - the ice dynamics term \$D\$. This is represented by the difference between 'model no D' and 'model with D' on Figure \ref{fig8}. \$D\$ is the only **simulated local** source or sink; in the natural system, \$D\$ produces ice roughness features such as rubble ice and pressure ridges, and lead timing, size, and distribution. Following any sea ice deformation, a certain amount of airborne snow will be removed to open water in the leads \citep{clemens2022} or stored in snowdrifts at the deformed ice roughness features \citep{liston2018,itkin2023}. During winter, the wind velocity is frequently above the blowing threshold value (7.7 m s\$^{-1}\$) following \cite{li\_pomeroy1997}, which provides the justification for our parametrization of \$D\$ as a sea ice deformation snow sink or source (see Section \ref{swe\_calc}). After melt onset, the snow grains are wet and no drifting snow is observed \citep[\textit{sensu}][]{pomeroy1997}. Following this principle, \$D\$ was set to zero in May after the last transect measurements.

\$D\$ remained small throughout the simulations, but its accumulated effect by 7 May (the last winter observation) was 10, 8, and >1 \% in Nloop, Sloop, and Runway, respectively. \$D\$ is likely large right after freeze-up; this is a period of thin ice with lots of deformation. More studies of this fast-changing period with thin ice are needed to understand what exactly is happening when the ice first forms. The importance of erosion for \$SWE\$ at MOSAiC was explored by \cite{wagner2022}, who gave estimates of erosion based on uncalibrated snowfall rates, wind speeds, and \$SWE\$ in Sloop and Nloop. While the magnitude of the combined snow sink by \cite{wagner2022} is similar to ours (53-68\%), their study could not differentiate between erosion and sublimation. Our study shows that \$D\$ was predominantly a sink in the case of Nloop and Runway and, after ridge formation in November, was occasionally a source in Sloop.

**NEW:**

On all three ice types, the strongest winter season sinks in our simulations were static and blowing snow sublimation (\$S\_{SS}\$ and\$S\_{BS}\$) which by 7 May cumulatively removed 67, 68, and 71 \% of \$SWE\$ from snowfall (\$P\_S\$) in Nloop, Sloop, and Runway, respectively. This is represented by the difference between 'precipitation' and 'model no D' on Figure \ref{fig8}. **The** magnitude of \$S\_{SS}\$ and \$S\_{BS}\$ depends on grain bonding, which is for \$S\_{SS}\$ determined by latent heat flux, while for \$S\_{BS}\$ wind speed, humidity and solar radiation **are the main factors \citep{liston2020}.** Note that, in this environment, if the blowing snow is not captured by **an** ice-topographic drift trap, or blown into an open lead, it blows perpetually and, in air that has a humidity deficit, it eventually sublimates completely away \ citep{tabler1975,liston sturm2004}. These \$\$ {\$\$}\$ and\$\$ {B\$}\$ values were about three times as large as in \cite{liston2020}. This is likely due to the specific weather during MOSAiC winter and location during the drift, including, generally low snowfall (\$P\_S\$) after freeze-up, frequent storms with high winds \citep{rinke2021}, and relatively high sea ice concentration \ citep{krumpen2021} with low near-surface relative humidity during winter. \$P\_S\$, \$S\_{SS}\$, and \$S\_{BS}\$ operate at synoptic temporal and length scales (on scales comparable to, e.g., 3 hours and 100 km), and were the same (or very similar for \$S\_{SS}\$ and \$S\_{BS}\$) for all ice types.

The differences in \$SWE\$ evolution on the three ice types were largely controlled by the ice (and snow) onset date, and the differences in the remaining wintertime snow sink or source - the ice dynamics term \$D\$. **These factors operate at much shorter local spatial scales, e.g., 10 m.** This is represented by the difference between 'model no D' and 'model with D' on Figure \ref{fig8}. \$D\$ is the only **local simulated** source or sink; in the natural system, \$D\$ produces ice roughness features such as rubble ice and pressure ridges, and lead timing, size, and distribution. Following

any sea ice deformation, a certain amount of airborne snow will be removed to open water in the leads  $\citep{clemens2022}$  or stored in snowdrifts at the deformed ice roughness features  $\citep{liston2018,itkin2023}$ . During winter, the wind velocity is frequently above the blowing threshold value (7.7 m s\$^{-1}\$) following  $\cite{li\_pomeroy1997}$ , which provides the justification for our parametrization of \$D\$ as a sea ice deformation snow sink or source (see Section  $\citep{swe\_calc}$ ). After melt onset, the snow grains are wet and no drifting snow is observed  $\citep{structit}$  textit{sensu}][]{pomeroy1997}. Following this principle, \$D\$ was set to zero in May after the last transect measurements.

\$D\$ remained small throughout the simulations, but its accumulated effect by 7 May (the last winter observation) was **15**, 8, and **<1** \% in Nloop, Sloop, and Runway, respectively. \$D\$ is likely large right after freeze-up; this is a period of thin ice with **frequent** deformation. More studies of this fast-changing period with thin ice are needed to understand what exactly is happening when the ice first forms. The importance of erosion for \$SWE\$ at MOSAiC was explored by \cite{wagner2022}, who gave estimates of erosion based on uncalibrated snowfall rates, wind speeds, and \$SWE\$ in Sloop and Nloop. While the magnitude of the combined snow sink by \cite{wagner2022} is similar to ours (53-68\%), their study could not differentiate between erosion-deposition and sublimation. Our study shows that \$D\$ was predominantly a sink (**erosion**) in the case of Nloop and Runway and, after ridge formation in November, was occasionally a source (**deposition**) in Sloop.

**Issue 3**

This section was modified significantly in the revision and the reviewer is right that the use of the 3-day running mean was still not explicitly addressed. For clarity we now add the statement: 'The 3-day running mean is used here to take into account the heat capacity of the freezing water column, opposed to freezing the ice surface.'

**Issue 4**

The Magnaprobe and snow stake snow depth observations are used in combination with snow densities from snow pits to calculate observed SWE. This is now explained at the end of Section 4.1, right after the conversion equation is introduced:

Because SnowModel-LG operates in \$SWE\$, while the MOSAiC observations provide \$h\_s\$ (Section \ref{sec\_snowdepth}) and \$\rho\_s\$ (Section \ref{sec\_snowdens} and specifically Equation \ref{eq\_rho} with continuous values for \$\rho\_s\$), any comparison can be made by using Equation \ref{eq\_hs} which determines the relationship between the three variables.

We realize that the mentioning of 'SWE cylinder' and 'SWE observations' can be confusing for the reader. We decided to replace the word 'SWE cylinder' with 'snow core sampler' in this manuscript.

Also all minor comments by this reviewer were addressed. In addition, we did some minor technical (language) corrections in the text and figures. All of them are tracked in the 'difference' PDF file.